# Achieving Fairness in Multi-Agent MDP Using Reinforcement Learning

**Peizhong Ju**
Department of ECE
The Ohio State University
Columbus, OH 43210, USA
ju.171@osu.edu

**Arnob Ghosh**
ECE Department
New Jersey Institute of Technology
Newark, NJ 07102, USA
arnob.ghosh@njit.edu

**Ness B. Shroff**
Department of ECE and CSE
The Ohio State University
Columbus, OH 43210, USA
shroff.11@osu.edu

## Abstract

Fairness plays a crucial role in various multi-agent systems (e.g., communication networks, financial markets, etc.). Many multi-agent dynamical interactions can be cast as Markov Decision Processes (MDPs). While existing research has focused on studying fairness in *known* environments, provably efficient exploration of fairness in such systems for *unknown* environments remains open. In this paper, we propose a Reinforcement Learning (RL) approach to achieve fairness in multi-agent finite-horizon episodic MDPs. Instead of maximizing the sum of individual agents' value functions, we introduce a fairness function that ensures equitable rewards across agents. Since the classical Bellman's equation does not hold when the sum of individual value functions is not maximized, we cannot use traditional approaches. Instead, in order to explore, we maintain a confidence bound of the unknown environment and then propose an online convex optimization based approach to obtain a policy constrained to this confidence region. We show that such an approach achieves sub-linear regret in terms of the number of episodes. Additionally, we provide a probably approximately correct (PAC) guarantee based on the obtained regret bound. We also propose an offline RL algorithm and bound the optimality gap with respect to the optimal fair solution. To mitigate computational complexity, we introduce a policy-gradient type method for the fair objective. Simulation experiments also demonstrate the efficacy of our approach.

## 1 Introduction

In classical Markov Decision Processes (MDPs), the primary objective is to find a policy that maximizes the reward obtained by a single agent over the course of an episode. However, in numerous real-world applications, decisions made by an agent can have an impact on multiple agents or entities. For instance, in a wireless network scenario, each device aims to maximize its own throughput by increasing its transmission power. However, higher transmission power can lead to interference issues for neighboring terminals. Similarly, consider a situation where two jobs are competing for a single machine; selecting one job results in a penalty or delay for the other job. The sequential decision-making process as in the above examples can be cast as a multi-agent episodic MDP where the central decision-maker seeks to obtain the best policy for multiple users or agents over a time horizon. Each user or agent achieves a reward (potentially different) based on the state and action.

Before delving into the concept of an optimal policy, it is necessary to address what constitutes an optimal policy in the given context. While a particular policy may be good for one agent, it may not be the best choice for another agent. A naive approach could be to maximize the aggregate value functions across all agents, thereby reducing the problem to a classical MDP. However, such an approach may not be considered "fair" for all agents involved. To illustrate this, consider a scenario where two jobs are competing for a single machine. If one job offers a higher reward, a

central controller that focuses solely on maximizing the aggregate reward may allocate the machine exclusively to the job with the higher reward, causing the job with the lower reward to remain in a waiting state indefinitely. In this paper, our objective is to identify fair decision-making strategies for multi-agent MDP problems, ensuring that all agents are treated equitably.

Drawing inspiration from well-known fairness principles (Arrow, 1965; Pratt, 1978; Atkinson et al., 1970), we establish a formalization of fairness as a function of the individual value function of agents. Specifically, we concentrate on $\alpha$-fairness, which encompasses both egalitarian or max-min fairness (when $\alpha \to \infty$) and proportional fairness (when $\alpha = 1$). The parameter $\alpha$ allows us to adjust the level of fairness desired. To illustrate this concept, let's consider our example of two jobs with different rewards competing for the same machine. Proportional fairness dictates that the machine should be accessed with equal probability by both the low-reward and high-reward jobs. Conversely, max-min fairness suggests that the job with the higher reward should access the machine with a probability that is inversely proportional to its reward.

In this work, we seek to determine the policy that maximizes the $\alpha$-fairness value of the individual value functions of an MDP. Considering that the knowledge of the environment is usually unknown beforehand in real-world applications, we consider a Reinforcement Learning (RL)-based approach. However, a significant challenge of non-linearity arises since the central controller is not optimizing the sum of the individual value functions, rendering the classical Bellman equation inapplicable. Consequently, conventional techniques such as value-iteration-based or policy-gradient-based approaches cannot be directly employed. To evaluate an online algorithm, regret is a widely used metric that measures the cumulative performance gap between the online solution in each episode and the optimal solution. While there are algorithms that provide good empirical performance, they do not provide any regret guarantee. Therefore, we aim to develop an algorithm that exhibits sub-linear regret with respect to the $\alpha$-fair solution. Further, since generating new data is costly or impossible for some applications, we also seek to develop a provably-efficient offline fair RL algorithm, i.e., an algorithm that requires no real-time new data. In short, we seek to answer–

*Can we attain a fair RL algorithm with sub-linear regret for multi-agent MDP? Can we develop a provably-efficient fair offline RL algorithm?*

**Our Contributions**: We summarize our contributions in the following:

- We show that our proposed algorithm achieves $\tilde{\mathcal{O}}\left(C_F(H^2NS\sqrt{AK})\right)$ regret where $H$ is the length of the horizon of each episode, $S$ is the cardinality of state space, $A$ is the cardinality of the action space, and $K$ is the number of episodes. $C_F$ is a parameter determined by the types/parameters of the fairness function.

- *This is the first sub-linear regret bound for the $\alpha$-fairness function in MDP.* We achieve the result by proposing an optimism-based convex optimization framework using state-action occupancy measures. In our algorithm, we use confidence bounds to quantify the error of the estimated reward and transition probability, with which we relax the constraints on possible values of the state-action occupancy measures to encourage exploration. With any convex optimization solver, our proposed algorithm can be solved efficiently in polynomial time. In order to address large state-space problems, we also develop an efficient policy-gradient-based approach that caters to the function approximation setup.

- We also propose a pessimistic version of the optimization problem and establish the theoretical guarantees for the offline fair RL setup. In particular, we construct an MDP with a reward function based on the available data such that the value function of the constructed MDP is a lower bound of the actual value function for the same policy with high probability. The policy is obtained by solving the convex optimization problem using the occupancy measure on the constructed MDP. We show that the sub-optimality gap of our policy depends on the intrinsic uncertainty multiplied by $C_F$. *This is the first result with a theoretical bound for the offline fair RL setup.*

## 2 RELATED WORK

**Fairness in resource allocation**: Fairness in traditional resource allocation setup has been well studied (Mo & Walrand, 2000; Kelly et al., 1998; Lin et al., 2006). RL-based fair resource allocation decision-making has also been considered for resource allocation (Chen et al., 2021; Hao et al., 2023; Jain et al., 2017; Cui et al., 2019). However, theoretical guarantees have not been provided.

**Fairness in MDP/RL**: Joseph et al. (2016); Liu et al. (2017) considered individual fairness criterion which stipulates that an RL system should make similar decisions for similar individuals. Huang et al. (2022); Schumann et al. (2019); Wen et al. (2021) considered a group fairness notion where the main focus is on policy is fair to a group of users (refer to Gajane et al. (2022) for details). Jiang & Lu (2019) proposed an approach where they perturbed the reward to make it fair across the users. In contrast to the above, our setup is different as we seek to achieve fairness in terms of value functions (i.e., long-term return) of different agents.

Hossain et al. (2021); Bistritz et al. (2020); Barman et al. (2022); Patil et al. (2021) considered Nash social welfare (or, proportional fairness) and other fairness notions in multi-armed bandit setup. Do et al. (2022) also considered fairness in the contextual bandit setup. However, we consider an RL setup instead of a bandit setup. The algorithms designed for bandit setup can not be readily extended to the MDP setup. Further, we consider the generic $\alpha$ fairness concept rather than the proportional-fairness concept. Zimmer et al. (2021); Siddique et al. (2020) considered $\alpha$-fairness and Gini social welfare metrics. However, the regret bounds have not been provided there. Finally, we provide a fair algorithm in an offline RL setup, which has not been considered in most of the RL literature.

The closest to our work is Mandal & Gan (2022) which adopted a welfare-based axiomatic approach and showed regret bound for Nash social welfare, and max-min fairness. In contrast, we considered the $\alpha$-based fairness metric and showed regret bound for the generic value of $\alpha$. Unlike in Mandal & Gan (2022), our approach admits efficient computation. Further, we provided the PAC guarantee and developed an algorithm for offline fair RL with a theoretical guarantee. Finally, we also developed a policy-gradient-based algorithm that is applicable to large state space as well.

**Convex and multi-objective MDP**: Cheung (2019) obtained regret bound for a specific non-linear function of the objectives. Tarbouriech et al. (2020) considered an MDP setup with a convex objective for infinite-horizon setup. Brantley et al. (2020) considered an episodic MDP setting where the objective is to maximize a concave function of an individual value function. Unlike all the above works, we consider a setup where the objective is to achieve fairness among multiple agents. Naturally, the above papers did not consider the effect of various fairness metrics on the agents. Further, the proof techniques and algorithms also rely on the Lipschitz property, however, the fairness function may not be Lipschitz (e.g., $\alpha$ fairness), hence, the proof techniques and the regret bound are also different. Furthermore, we consider offline setup unlike all the above papers. For detailed literature review, please see Appendix A.

## 3 BACKGROUND: TYPES OF FAIRNESS IN RESOURCE ALLOCATION

Fairness in resource allocation in multi-agent systems (especially in networks) has been extensively studied (Lin & Shroff, 2005; 2004; Eryilmaz & Srikant, 2007; Neely et al., 2008). Specifically, in resource allocation, a feasible solution is any vector $\boldsymbol{x} := [x_1 \ x_2 \ \cdots \ x_N] \in \mathcal{F} \subseteq \mathbb{R}_+^N$ where $N$ denotes the number of agents, $x_i$ denotes the allocated resource to each agent, and $\mathcal{F}$ denotes a feasible set determined by some constraints. A fair objective is to allocate resources while maintaining some kind of fairness. As described in Mo & Walrand (2000), the following are some standard definitions of fairness.

**Proportional Fairness**: A solution $\boldsymbol{x}^*$ is proportional fair when it is feasible and for any other feasible solution $\boldsymbol{x} \in \mathcal{F}$, the aggregate of proportional change is non-positive:

$$\sum_{i=1}^N \frac{x_i - x_i^*}{x_i^*} \leq 0. \tag{1}$$

In particular, for all other allocations, the sum of proportional rate changes with respect to $x^*$ is non-positive. Proportional fairness is widely used in network applications such as scheduling.

**Max-min fairness**: Max-min fairness wants to get a feasible solution that maximizes the minimum resources of all agents, i.e., $\max_{x \in \mathcal{F}} \min_i x_i$. For this solution, no agent can get more resources without sacrificing another agent's resources.

**$(p, \alpha)$-proportional fairness**: Let $p = (p_1, \cdots, p_N)$ and $\alpha$ be positive numbers. A solution $\boldsymbol{x}^*$ is $(p, \alpha)$-proportionally fair when it is feasible and for any other feasible solution $\boldsymbol{x} \in \mathcal{F}$, we have

$$\sum_{i=1}^N p_i \frac{x_i - x_i^*}{x_i^{*\alpha}} \leq 0. \tag{2}$$

When $p = (1, \cdots, 1)$ and $\alpha = 1$, Eq. (2) reduces to Eq. (1), i.e., the proportionally fair solution. Besides, by Corollary 2 of Mo & Walrand (2000), the solution of $(p, \alpha)$-proportional fair approaches the one of max-min fair as $\alpha \to \infty$. By Lemma 2 of Mo & Walrand (2000), the solution that achieves $(p, \alpha)$-proportional fairness can be solved by

$$\max_{\boldsymbol{x}} \sum_i p_i f_\alpha(x_i), \text{ where } f_\alpha(x) = \begin{cases} \log x, & \text{if } \alpha = 1 \\ (1-\alpha)^{-1} x^{1-\alpha}, & \text{other positive } \alpha. \end{cases}$$

When $p_i = 1$ for all $i$, we denote that by $\alpha$-fairness in short, which is widely studied in the networking literature (Lan et al., 2010). Note that $f_\alpha$ is a monotonic increasing function, and concave.

## 4 SYSTEM MODEL

**Multi-agent Finite Horizon MDPs.** Let $\mathcal{M} = (N, \mathcal{S}, \mathcal{A}, r, p, s_1, H)$ be the finite-horizon MDP, where $N$ denotes the number of agents, $\mathcal{A}$ denotes the action space with cardinality $A$, $\mathcal{S}$ denotes the state space with cardinality $S$, and $H$ is a positive integer that denotes the horizon length. At time $h = 1, 2, \cdots, H$, we let $r_{h,(i)}(s, a)$ denote the non-stationary immediate reward for the $i$-th agent when action is $a \in \mathcal{A}$ at state $s \in \mathcal{S}$. The transition probability is denoted by $p_h(s'|s, a)$. Note that this setup can be easily extended to the scenario where agents are also part of the MDP by letting $\mathcal{A}$ denote the joint action space of all agents.

### 4.1 VALUE FUNCTION AND FAIRNESS

The state-action value function of agent $i$ is defined as

$$Q_{h,(i)}^\pi(s, a) := r_{h,(i)}(s, a) + \mathbb{E}\Big[ \sum_{l=h+1}^H r_{l,(i)}(s_l, a_l)|s_h = s, a_h = a, \pi, p\Big].$$

The $i$-th agent's value function is defined as $V_{h,(i)}^\pi(s) := \sum_{a \in \mathcal{A}} \pi_h(a|s) Q_{h,(i)}^\pi(s, a)$.

To achieve fairness among each agent's return, we optimize a different global value function (instead of $V_1^{\pi,\text{sum}}(s)$ that sums up each agent's return):

$$V_1^{\pi,F}(s) := F(V_{1,(1)}^\pi(s), V_{1,(2)}^\pi(s), \cdots, V_{1,(N)}^\pi(s)), \tag{3}$$

where $F$ is some function of every agent's return that can be chosen for fairness. Similar to the fairness objective used in resource allocation literature (Mo & Walrand, 2000), we consider the following three possible options of $F$:

$$F_{\text{max-min}} = \min\{V_{1,(1)}^\pi(s), V_{1,(2)}^\pi(s), \cdots, V_{1,(N)}^\pi(s)\} \quad \text{(max-min fairness)}, \tag{4}$$

$$F_{\text{proportional}} = \sum_{i=1}^N \log V_{1,(i)}^\pi(s) \quad \text{(proportional fairness)}, \tag{5}$$

$$F_\alpha = \sum_{i=1}^N \frac{1}{1-\alpha} V_{1,(i)}^\pi(s)^{1-\alpha} \quad (\alpha \text{ fairness where } \alpha > 0), \tag{6}$$

Note that we have adopted the $\alpha$-fairness in resource allocation to the $\alpha$-fairness in value functions across the agents. Chen et al. (2021) also formalized fairness among value functions in network applications. Recently, Zhang et al. (2022) also adopted $\alpha$-fairness to federated learning setup. In the rest of this paper, we sometimes remove the superscript $F$ in $V_1^{\pi,F}(s)$ for ease of notation.

**Remarks and Connections:** From (1), maximizing proportional fairness in the value function means that the average relative value function is maximized. In particular, at any other policy, the average relative value function across the agents would be reduced compared to the proportionally fair maximizing policy. Hossain et al. (2021); Mandal & Gan (2022) maximize the product (contrast to the sum) of each agent's value function $\prod_{i=1}^N V_{1,(i)}^\pi(s)$ (also, known as Nash social welfare). By taking the logarithm on the product, it is equivalent to $F_{\text{proportional}}$ (Kelly, 1997) in Eq. (5) in our case. Mandal & Gan (2022) has a regret bound for Nash social welfare. Even though the proportional-fair solution is Nash social welfare solution, the regret bound for the Nash social welfare and for the proportional fair case is not comparable. For example, their bound scales $O(H^N)$, whereas our regret bound scales as $O(NH^2)$ (shown later in this paper). The proof technique is also different.

When $\alpha = 0$, we recover the utilitarian social welfare where the objective is to maximize the sum of the value functions. On the other hand, $\alpha = \infty$ refers to max-min fairness in value functions. By tuning $\alpha$, one can achieve different fairness metrics.

**Social Impact:** One can view different types of fairness as achieving different goals in society. For example, max-min fairness aims to make decisions that are beneficial to the weakest users.

## 4.2 PERFORMANCE EVALUATION

Define the optimal *fair* value function corresponding to the optimal policy as

$$V_1^*(s) = \sup_\pi V_1^\pi(s).$$

The central controller does not know either the probability or the rewards. Rather, it selects a policy $\pi_k$ for $k \in [K]$ episode. Without loss of generality, we assume that the initial state $s_1$ for all episodes is the same and fixed. If the initial state $s_1$ is drawn from some distribution, then we can construct an artificial initial state $s_0$ that is fixed for all episodes, and the distribution of the actual initial state $s_1$ determines the transition probability $p_0(s_1|s_0, a)$. We consider the *bandit-feedback setup*, i.e., the central controller can only observe the rewards (of all the agents) corresponding to the encountered state-action pair (Agarwal et al., 2011; Dani et al., 2008). We assume the following for the reward:

**Assumption 1.** *The noisy observation of the immediate reward is a random variable $\hat{r}_{h,(i)}(s, a)$, which is in the range $[\frac{\epsilon}{H}, 1]$ almost surely where $\epsilon$ is some positive real number. The mean value of the noisy observation is equal to the true immediate reward, i.e., $\mathbb{E}\,\hat{r}_{h,(i)}(s, a) = r_{h,(i)}(s, a)$.*

**Remark 1.** *We need $\hat{r} \geq \frac{\epsilon}{H}$ because this guarantee $V_{1,(i)}^\pi(s_1) \geq \epsilon > 0$ which ensures that Eqs. (5) and (6) are finite. Also, this makes the functions Lipschitz continuous everywhere. We characterize the impact of $\epsilon$ on the regret/suboptimality bound in Theorem 1.*

We are interested in minimizing the regret $\mathsf{Reg}(K)$ over finite time horizon $K$, given by

$$\mathsf{Reg}(K) \coloneqq \sum_{k=1}^K \left(V_1^*(s_1) - V_1^{\pi_k}(s_1)\right).$$

The regret characterizes the cumulative sum of the difference at each episode $k = 1, 2, \cdots, K$ between the fair value function and the optimal fair value function.

## 5 ONLINE FAIR MARL

### 5.1 OPTIMAL POLICY WITH COMPLETE INFORMATION

Before we characterize the algorithm when the MDP parameters are unknown, we start from the ideal situation where all parameters of the MDP are known, i.e., complete information. The insight will help us to develop an algorithm for the challenging scenario when the parameters are unknown.

For the classical objective that maximizes the sum of all agents' returns, the optimal return and policy can be efficiently calculated by backward induction that utilizes the Bellman equation, i.e,

$$V_h^{*,\text{sum}}(s) = \max_{a \in \mathcal{A}} \left\{ \sum_{i=1}^N r_{h,(i)}(s, a) + \sum_{s' \in \mathcal{S}} p_h(s'|s, a) V_{h+1}^{*,\text{sum}}(s') \right\}, \tag{7}$$

where $V_{H+1}^{*,\text{sum}}(s) = 0$ for all $s \in \mathcal{S}$. The reason for Eq. (7) is that maximize $\sum_{i=1}^N V_{h,(i)}^\pi(s)$ is equivalent to solving another single-agent MDP with immediate reward equal to $\sum_{i=1}^N r_{h,(i)}(s, a)$.

In contrast, such a convenience no longer exists for the fairness objective since Eq. (7) relies on the linearity of $V_h^{\pi,\text{sum}}(s)$ w.r.t. $V_{h,(i)}^\pi(s)$. To solve this problem, we alternatively use an occupancy-measure-based approach which is inspired by Efroni et al. (2020). Define the occupancy measure

$$q_h^\pi(s, a; p) \coloneqq \Pr\{s_h = s, a_h = a \mid s_1, p, \pi\}. \tag{8}$$

The occupancy measure defined by Eq. (8) represents the frequency of the appearance for each state-action pair under the policy $\pi$ on the environment transition probability $p$. We will omit $p$ in the notation $q_h^\pi(s, a; p)$ when the context is clear.

With this definition, each agent's return can be written as a linear function w.r.t. $q_h^\pi(s, a)$, i.e.,

$$V_{1,(i)}^\pi(s_1) = \sum_{s,a,h} r_{h,(i)}(s, a) \cdot q_h^\pi(s, a). \tag{9}$$

Then we can solve a convex optimization of $q$ (we use $[\cdot]_{i=1,2,\cdots,N}$ to denote $N$ inputs of $F(\cdot)$):

$$\max_{q \in \mathcal{Q}} \quad F\left( \left[\sum_{s,a,h} r_{h,(i)}(s,a) \cdot q_h^\pi(s,a)\right]_{i=1,2,\cdots,N} \right) \quad (\text{i.e., } \max_{q \in \mathcal{Q}} V_1^{\pi,F}(s_1)), \qquad (10)$$

where $\mathcal{Q}$ is a set of linear constraints on $q$ to make sure $q$ is a legit occupancy measure with the transition probability $p$ and initial state $s_1$ (details in Appendix C). Since (10) is a convex optimization (proof in Lemma 6 in Appendix B), and thus can be solved efficiently in polynomial time. After we get the occupancy measure $q$, the corresponding policy can be calculated by $\pi_h(a|s) = \frac{q_h(s,a)}{\sum_{a'} q_h(s,a')}$.

## 5.2 ONLINE ALGORITHM WITH UNKNOWN ENVIRONMENT

To construct an online algorithm under the bandit-feedback, a straightforward idea is using the empirical average $\bar{p}, \bar{r}$ (precisely defined in Eqs. (31) and (30) in Appendix C) of the unknown transition probability $p$ and reward $r$ to replace the precise ones in (10). Due to the imprecision of the empirical average, a common strategy is to introduce some confidence interval to balance exploration and exploitation as done in Efroni et al. (2020). Here we briefly show the algorithm. More details are in Appendix C.

Define the confidence interval for $\bar{p}_h^{k-1}(s'|s,a)$ as $\beta_{h,k}^p(s,a,s')$ such that

$$\left| \bar{p}_h^{k-1}(s'|s,a) - \tilde{p}_h(s'|s,a) \right| \leq \beta_{h,k}^p(s,a,s'), \text{ for all } h \in [H-1], s,s' \in \mathcal{S}, a \in \mathcal{A}. \qquad (11)$$

Define the confidence interval for $\bar{r}_{h,(i)}^{k-1}(s,a)$ as $\beta_{h,k}^r(s,a)$ such that

$$\left| \tilde{r}_{h,(i)}(s,a) - \bar{r}_{h,(i)}^{k-1}(s,a) \right| \leq \beta_{h,k}^r(s,a), \text{ for all } i \in [N], h \in [H], s \in \mathcal{S}, a \in \mathcal{A}. \qquad (12)$$

After we get the confidence interval at the $h$-th step during the $k$-th iteration, we solve the following extended convex optimization:

$$\max_{z \in \mathcal{Z}} F\left( \left[\sum_{s,a,h,s'} \left( \bar{r}_{h,(i)}^{k-1}(s,a) + \beta_{h,k}^r(s,a) \right) \cdot z_h(s,a,s') \right]_{i=1,2,\cdots,N} \right), \qquad (13)$$

where $\mathcal{Z}$ is a set of constraints that ensures $z$ is a legit state-action-next-state occupancy measure given initial state $s_1$ (characterized by the set of probable transition probabilities, see Appendix C). Once we have solved $z$, we can recover the policy by $\pi_{k,h}(a|s) = \frac{\sum_{s'} z_h(s,a,s')}{\sum_{a',s'} z_h(s,a',s')}$. The whole algorithm is summarized in Algorithm 1. Mandal & Gan (2022) developed an algorithm using the state-action occupancy measure. However, the algorithm in Mandal & Gan (2022) relies on an optimization problem with infinite variables, which does not always have a polynomial solver. In contrast, our approach requires only finite variables and is more efficient.

---

**Algorithm 1** Online FairMARL

1: **for** $k = 1, 2, \cdots, K$ **do**
2:     Calculate the empirical average $\bar{p}_h^{k-1}(s'|s,a)$, and $\bar{r}_{h,(i)}^{k-1}(s,a)$.
3:     Calculate the confidence intervals $\beta_{h,k}^p(s,a,s')$ and $\beta_{h,k}^r(s,a)$.
4:     Compute policy $\pi_k$ by solving (13).
5:     Execute the policy $\pi_k$.
6: **end for**

---

**Theorem 1.** *With probability $1 - \delta$, we have*

$$\text{Reg}(K) = C_F \cdot \left( \tilde{\mathcal{O}}(H^2 N S \sqrt{AK}) + \tilde{\mathcal{O}}(H N^2 S^{3/2} A) + \tilde{\mathcal{O}}(H^2 N S^2 A) \right),$$

*where $C_F$ is a constant determined by the type of fairness. Specifically,*

$$C_F = \begin{cases} \epsilon^{-\alpha} & \text{when } F = F_\alpha \text{ ($\alpha$ fairness)} \\ \epsilon^{-1} & \text{when } F = F_{proportional} \text{ (proportional fairness)} \\ 1/N & \text{when } F = F_{max\text{-}min} \text{ (max-min fairness)} \end{cases}.$$

*The notation $\tilde{\mathcal{O}}(\cdot)$ ignores logarithm terms (such as $\log \frac{1}{\delta}$).*

Proof of Theorem 1 is in Appendix C. Here we provide a proof outline:

First, for the optimism choice of $q$ (i.e., $\pi_k$), $\tilde{r}$, $\tilde{p}$, we have $V_1^{\pi_k}(s_1; \tilde{r}, \tilde{p}) \geq V_1^*(s_1; r, p)$ when $r, p \in M_k$ (which happens with high probability). By the montonocity property of $F(\cdot)$, we thus have $\sum_{k=1}^K \left( V_1^{*,F}(s_1) - V_1^{\pi_k,F}(s_1) \right) \leq \sum_{k=1}^K \left( V_1^{\pi_k,F}(s_1; \tilde{r}, \tilde{p}) - V_1^{\pi_k,F}(s_1; r, p) \right)$.

Next, we need to bound $V_1^{\pi_k,F}(s_1; \tilde{r}, \tilde{p}) - V_1^{\pi_k,F}(s_1; r, p)$. However, unlike the traditional techniques, we can not use the standard value-difference lemma to bound the above as the Bellman's equation does not hold. Rather, we obtain the bound in two steps. First, we bound the above by $C_F \sum_{i=1}^N \left| V_{1,(i)}^{\pi_k}(s_1; \tilde{r}, \tilde{p}) - V_{1,(i)}^{\pi_k}(s_1; r, p) \right|$ or by $C_F \cdot N \max_{i \in [N]} \left| V_{1,(i)}^{\pi_k}(s_1; \tilde{r}, \tilde{p}) - V_{1,(i)}^{\pi_k}(s_1; r, p) \right|$. The value of $C_F$ is determined by the Lipchitz constant of the fairness objective function $F$ or the property of the max-min operator. Then we bound the individual differences $\left| V_{1,(i)}^{\pi_k}(s_1; \tilde{r}, \tilde{p}) - V_{1,(i)}^{\pi_k}(s_1; r, p) \right|$ using Azuma-Hoeffding inequality. The result of Theorem 1 then follows.

**Remark 2.** *For max-min fairness, the requirement $\hat{r} \geq \frac{\epsilon}{H}$ in Assumption 1 can be relaxed.*

To the best of our knowledge, this is the first sub-linear regret for $\alpha$-fair RL. When $\alpha = 0$, we recover the single-agent regret (scaled by $N$) as it is equivalent to the MDP where the reward is $\sum_i r_{h,(i)}$. The constant $C_F$ decreases as $\alpha$ increases. For the max-min fairness, our result matches that of Mandal & Gan (2022), although our algorithm is more efficient.

**From regret to PAC guarantee**: The probably approximately correct (PAC) guarantee shows how many samples are needed to find an $\varepsilon$-optimal policy $\pi$ satisfying $V_1^*(s_1) - V_1^\pi(s_1) \leq \varepsilon$ (Jin et al., 2018; Valiant, 1984). Similar to Section 3.1 in Jin et al. (2018), in order to get the probably approximately correct (PAC) guarantee from regret, we can randomly select $\pi = \pi_k$ for $k = 1, 2, \cdots, K$. We define such a policy as $\pi^{\text{mix}}$. However, since $V_1^\pi(s)$ is not linear w.r.t. the immediate reward $r$, generally $V_1^{\pi^{\text{mix}},F}(s) \neq \frac{1}{K} \sum_{k=1}^K V_1^{\pi_k,F}(s)$. Therefore, compared with Jin et al. (2018), some additional derivation is needed to achieve PAC guarantee from regret in our case. In particular, from Jensen's inequality (since $F$ is concave), $V_1^{\pi^{\text{mix}}}(s_1) \geq \frac{1}{K} \sum_{k=1}^K V_1^{\pi_k}(s_1)$. We obtain

**Theorem 2.** *To find $\varepsilon$-optimal policy, with high probability, it suffices to have $C$ number of samples where*
$$C = C_F \max \left\{ \tilde{\mathcal{O}}(H^5 N^2 S^2 A / \varepsilon^2), \ \tilde{\mathcal{O}}(H^3 N^4 S^3 A^2 / \varepsilon^2), \ \tilde{\mathcal{O}}(H^3 N^2 S^4 A^2 / \varepsilon^2) \right\}.$$

Proof of Theorem 2 is in Appendix D.

**Fair Online Policy Gradient.** In the proposed convex-optimization-based algorithm, the decision variable and the constraints scale with the cardinality of the state space. In order to develop an algorithm for large state space, generally function approximation-based approaches (e.g., using neural networks) are used to approximate the $Q$ function or value function. We can use policy-gradient methods that cater to such a function approximation-based approach. In particular, consider a trajectory $\tau = (s_h^\tau, a_h^\tau, \hat{r}_h^\tau)_{h=1,2,\cdots,H}$ where $\hat{r}_h^\tau = (\hat{r}_{h,(1)}^\tau, \cdots, \hat{r}_{h,(N)}^\tau)$ denotes the noisy observation of immediate reward for all agents. We define the return for the $i$-th agent as $R_{(i)}(\tau) \coloneqq \sum_{h=1}^H \hat{r}_{h,(i)}^\tau$. To calculate the gradient of the fair objective, we can apply the chain rule of $\nabla_{\boldsymbol{\theta}} F(\cdot)$. We use proportional fairness $F_{\text{proportional}}$ as an example of calculating the gradient:

$$\nabla_{\boldsymbol{\theta}} V_1^{\pi_{\boldsymbol{\theta}},F}(s_1) = \nabla_{\boldsymbol{\theta}} \sum_{i=1}^N \log V_{1,(i)}^{\pi_{\boldsymbol{\theta}}}(s_1) = \sum_{i=1}^N \frac{\partial \log(V_{1,(i)}^{\pi_{\boldsymbol{\theta}}}(s_1))}{\partial V_{1,(i)}^{\pi_{\boldsymbol{\theta}}}(s_1)} \nabla_{\boldsymbol{\theta}} V_{1,(i)}^{\pi_{\boldsymbol{\theta}}}(s_1) = \sum_{i=1}^N \frac{\nabla_{\boldsymbol{\theta}} V_{1,(i)}^{\pi_{\boldsymbol{\theta}}}(s_1)}{V_{1,(i)}^{\pi_{\boldsymbol{\theta}}}(s_1)}.$$

It is known that $\nabla_{\boldsymbol{\theta}} V_{1,(i)}^{\pi_{\boldsymbol{\theta}}}(s_1) = \mathbb{E}_\tau[R_{(i)}(\tau) \log \pi_{\boldsymbol{\theta}}(a_h^\tau | s_h^\tau)]$ and $V_{1,(i)}^{\pi_{\boldsymbol{\theta}}}(s_1) = \mathbb{E}_\tau[R_{(i)}(\tau)]$. By using the empirical average to replace $\mathbb{E}_\tau$, we can get an unbiased estimator of gradient w.r.t. $\boldsymbol{\theta}$ as follows:
$\boldsymbol{g}_{\text{proportional}} = \sum_{i=1}^N \frac{\sum_{\tau \in \mathcal{D}} \sum_{h=1}^H R_{(i)}(\tau) \nabla_{\boldsymbol{\theta}} \log \pi_{\boldsymbol{\theta}}(a_h^\tau | s_h^\tau)}{\sum_{\tau \in \mathcal{D}} R_{(i)}(\tau)}$.

For other types of fairness, we can use a similar method. The final expression of the gradient, the rest part of the algorithm, and other related details are in Appendix F. Note that we can extend this approach to the natural policy-gradient, actor-critic method, and baseline-based approach. Characterization of the convergence rate of the approaches are beyond the scope of this paper. Interested

readers can refer to Zhang et al. (2020); Agarwal et al. (2021); Mei et al. (2020) for convergence analysis of standard policy gradient.

# 6 OFFLINE FAIR MARL

As mentioned in the Introduction, we develop an offline algorithm because for some applications generating new data may not be feasible. In an offline setting, the learner is given a dataset and it needs to compute a policy only based on this given dataset. One cannot employ a policy and measure its return. Due to this difference, instead of optimism, pessimism is optimal for standard MDP (Xie et al., 2021; Jin et al., 2021b). We develop an offline fair algorithm and analyze its suboptimality gap. Before delving into the result, we need to have some assumptions about the data collection process.

**Assumption 2.** *The dataset* $\mathcal{D} = \{r_{h,(i)}^\tau, x_h^\tau, a_h^\tau\}_{h\in[H],\tau\in[K],i\in[N]}$ *is compliant with the underlying MDP, i.e.,* $\forall i$,

$$\mathbb{P}_{\mathcal{D}}(r_{h,(i)}^\tau = r_i', x_{h+1}^\tau = x'|\{(x_h^j, a_h^j)\}_{j=1}^\tau, \{(r_{h,(i)}^j, x_{h+1}^j)\}_{j=1}^{\tau-1})$$
$$=\mathbb{P}(r_{h,(i)}(s_h, a_h) = r_i', s_{h+1} = x'|s_h = x_h^\tau, a_h = a_h^\tau).$$

The above assumption is satisfied when the data is collected by interacting with the environment and the policy is only updated at the end of an episode. Jin et al. (2021b) also uses a similar assumption. Similar to the online algorithm, we denote the empirical estimation $\overline{p}$ and $\overline{r}$ on $p$ and $r$, respectively, for the dataset $\mathcal{D}$. We first define the uncertainty quantizer for the data set which we use to construct MDP with pessimistic reward.

**Definition 1.** *We define the set* $\mathcal{E}$ *as the $\delta$-uncenrtainty quantifier with respect to the dataset* $\mathcal{D}$ *as–*

$$\mathcal{E} = \{b_h^r(s,a,\delta), b_h^p(s,a,s',\delta) : \left|\overline{r}_{h,(i)}(s,a) - r_{h,(i)}(s,a)\right| \le b_h^r(s,a,\delta)$$
$$|\overline{p}_h(s'|s,a) - p_h(s'|s,a)| \le b_h^p(s,a,s',\delta), \forall i,s,a,h\}$$

*such that* $\mathbb{P}_{\mathcal{D}}(\mathcal{E}) \ge 1 - \delta$.

The values of $b_h^r(s,a,\delta), b_h^p(s,a,s',\delta)$ are given in Appendix E. They are related to $\beta_{h,k}^r$, and $\beta_{h,k}^p$. The only difference is that the empirical estimation now depends on the dataset rather than the obtained information till episode $k$ in the online version.

We can show that with probability $1 - \delta$, for any $V_{h,(i)}$

$$|\mathbb{P}_h[V_{h,(i)}] - \overline{\mathbb{P}}_h[V_{h,(i)}](s,a)| = \sum_{s'} |(\overline{p}_h(s'|s,a) - p_h(s'|s,a))V_{h,(i)}(s')| \le H \sum_{s'} b_h^p(s,a,s',\delta).$$

We then define the pessimistic reward $\underline{r}_{h,(i)}$ as

$$\underline{r}_{h,(i)}(s,a) := \overline{r}_{h,(i)}(s,a) - b_h^r(s,a,\delta) - H \sum_{s'} b_h^p(s,a,s',\delta).$$

Note that we have also subtracted $H \sum_{s'} b_h^p(s,a,s')$ in order to ensure the value function attained for the MDP with reward $\underline{r}_{h,(i)}$ and empirical probability $\overline{p}_h$ is less than the value function corresponding to the original MDP parameters for the same policy, i.e., ensure pessimism. Note that similar pessimistic estimate is also used in constrained MDP setup (Liu et al., 2021). However, our proof and algorithms are different as we consider a fair objective.

To bound the suboptimality, we need an additional assumption that each agent's return under pessimistic reward $\underline{r}$ should be positive and shouldn't be too small. Specifically, we need the following assumption.

**Assumption 3.** $V_{1,(i)}(s_1, \underline{r}, \overline{p}) \ge \epsilon$ *for all* $i$.

The above assumption is required to apply the Lipschitz continuous property (see Lemma 5 in Appendix B). If $\underline{r}_{h,(i)}(s,a) \ge \epsilon/H$ for every $h,i,s,a$, then the above Assumption 3 is trivially satisfied. Also, our analysis would go through using a slightly larger $\underline{r}$ since the true reward value is greater than or equal to $\epsilon/H$. In particular, we can set $\underline{r}_{h,(i)}(s,a) = \max\{\overline{r}_{h,(i)}(s,a) - b_h^r(s,a,\delta), \epsilon/H\} - H \sum_{s'} b_h^p(s,a,s')$. Hence, it is clear that Assumption 3 is more likely to hold when the uncertainty

on the estimation of $p$ in the offline data is small. This is reasonable because when the uncertainty is high, it is unlikely to bound the regret, especially since some fair objectives $F$ are unbounded when any agent's return is near $0$.

Our proposed offline algorithm is solving the following convex optimization:

$$\max_{q \in \overline{\mathcal{Q}}} \quad F\big(\big[\sum_{h,s,a} \underline{r}_{h,(i)}(s,a)q_h(s,a)\big]_{i=1,2,\cdots,N}\big), \tag{14}$$

where $\overline{\mathcal{Q}}$ is the same as $\mathcal{Q}$ in (10) but with $\overline{p}$ instead of $p$. Similar to the online algorithm, we still use the occupancy measure $q$ to construct a convex optimization problem. However, compared with the online version, a key difference is that we use a pessimistic reward instead of the optimistic reward in the objective. Further, the MDP is based on the empirical probability $\overline{p}$, unlike the online setup where we allow the probability to take value within the confidence interval.

### 6.1 PERFORMANCE GUARANTEE OF THE OFFLINE ALGORITHM

We denote the solution of Eq. (14) as $\hat{q}$ and the corresponding policy as $\hat{\pi}$. The suboptimality of any policy $\pi$ is defined by

$$\mathsf{SubOpt}(\pi; s) := V_1^{\pi^*}(s; r, p) - V_1^{\pi}(s; r, p).$$

**Theorem 3.** *Given offline data $\mathcal{D}$, with probability $1 - \delta$*

$$\mathsf{SubOpt}(\hat{\pi}; s_1) \le 2NC_F \mathbb{E}_{\pi^*}[\underbrace{\sum_h ((b_h^r(s_h, a_h, \delta) + H\sum_{s'} b_h^p(s_h, a_h, s', \delta)))}_{\mathrm{Intrinsic-Uncertainty}}]. \tag{15}$$

To the best of our knowledge, this is the first offline RL result for the $\alpha$-fairness function. In the standard single-agent MDP, the result also depends on the $\delta$ uncertainty quantifier term and intrinsic uncertainty term that constitutes information theoretic lower limit on optimality-gap (Jin et al., 2021b). Here, it is scaled by $N$ and $C_F$. $C_F$ is the Lipschitz constant which depends on $\alpha$-fairness function. Further, if the dataset $\mathcal{D}$ has good coverage over the optimal policy, then the $\mathrm{Intrinsic-Uncertainty}$ term is small.

Proof of Theorem 3 is in Appendix E. Here we provide a proof sketch: We have

$$\mathsf{SubOpt}(\hat{\pi}; s) = \underbrace{\left(V_1^{\pi^*, F}(s; r, p) - V_1^{\pi^*, F}(s; \underline{r}, \overline{p})\right)}_{\mathrm{Term\ 1}} + \underbrace{\left(V_1^{\pi^*, F}(s; \underline{r}, \overline{p}) - V_1^{\hat{\pi}, F}(s; \underline{r}, \overline{p})\right)}_{\mathrm{Term\ 2}}$$
$$+ \underbrace{\left(V_1^{\hat{\pi}, F}(s; \underline{r}, \overline{p}) - V_1^{\hat{\pi}, F}(s; r, p)\right)}_{\mathrm{Term\ 3}}. \tag{16}$$

Term 2 of Eq. (16) is non-positive because $\hat{\pi}$ is the solution of Eq. (14). In standard offline RL literature (Jin et al., 2021b; Xie et al., 2021), Term 3 of Eq. (16) is non-positive because of the pessimism which is proved using Bellman's property. However, since Bellman's property does not hold, we cannot use the standard technique. Rather, we use the fact that $F(\cdot)$ is monotone increasing w.r.t. $r$ to show Term 3 is non-positive. For Term 1, we use the Lipschitz property of $F(\cdot)$ to show

$$\left(V_1^{\pi^*, F}(s; r, p) - V_1^{\pi^*, F}(s; \underline{r}, \overline{p})\right) \le \sum_{i=1}^N C_F |V_{1,(i)}^{\pi^*}(s; r, p) - V_{1,(i)}^{\pi^*}(s; \underline{r}, \overline{p})|. \tag{17}$$

The right-hand side then becomes differences of individual value functions and can be bounded by the Value-difference lemma.

## 7 NUMERICAL RESULTS

We have conducted experiments on randomly generated MDP environments to verify our online and offline algorithms. Please see Appendix G for details.

## 8 CONCLUSION AND FUTURE WORK

In this paper, we develop convex-optimization-based algorithms for both the online and offline fair RL with provable performance guarantee. Potential future directions include studying decentralized fair MARL algorithms and other policy gradient methods along with their convergence. Developing provably-efficient fair RL algorithms beyond tabular setup constitutes a future research direction.

## 9 ACKNOWLEDGEMENT

This work has been supported in part by NSF grants: CNS-2312836, CNS-2223452, CNS-2225561, CNS-2112471, CNS-2106933, a grant from the Army Research Office: W911NF-21-1-0244, and was sponsored by the Army Research Laboratory under Cooperative Agreement Number W911NF-23-2-0225. The views and conclusions contained in this document are those of the authors and should not be interpreted as representing the official policies, either expressed or implied, of the Army Research Laboratory or the U.S. Government. The U.S. Government is authorized to reproduce and distribute reprints for Government purposes notwithstanding any copyright notation herein. AG was also partly supported by the NJIT startup grant 172884.

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

# Supplemental Material

## A RELATED WORKS IN DETAIL

**Fairness in resource allocation**: Fairness in traditional resource allocation setup has been well studied (Mo & Walrand, 2000; Kelly et al., 1998; Lin et al., 2006). RL-based fair resource allocation decision-making has also been considered for resource allocation (Chen et al., 2021; Hao et al., 2023; Jain et al., 2017; Cui et al., 2019). However, theoretical guarantees have not been provided.

**Fairness in MDP/RL**: Zhang & Shah (2014) proposed max-min fairness in MDP, however, the learning component has not been considered. Joseph et al. (2016); Liu et al. (2017) considered individual fairness criterion which stipulates that an RL system should make similar decisions for similar individuals or the worst action should not be selected compared to a better one. Huang et al. (2022); Schumann et al. (2019); Wen et al. (2021) considered a group fairness notion where the main focus is on policy is fair to a group of users (refer to Gajane et al. (2022) for details). Deng et al. (2022); Metevier et al. (2019) considered an approach where fairness is modeled as a constraint to be satisfied. (Jiang & Lu, 2019) proposed an approach where they perturbed the reward to make it fair across the users. In contrast to the above, our setup is different as we seek to achieve fairness in terms of value functions (i.e., long-term return) of different agents.

Zimmer et al. (2021); Siddique et al. (2020) considered Gini-fairness across the value functions of the multiple agents, while we focus on different fairness metrics. Besides, the regret bounds have not been provided there. Hossain et al. (2021); Bistritz et al. (2020); Barman et al. (2022); Patil et al. (2021) considered Nash social welfare (or, proportional fairness) and other fairness notions in multi-armed bandit setup. Do et al. (2022) also considered fairness in the contextual bandit setup. However, we consider an RL setup instead of a bandit setup. The algorithms designed for bandit setup can not be readily extended to the MDP setup. Further, we consider the generic $\alpha$ fairness concept rather than the proportional-fairness concept. Finally, we provide a fair algorithm in an offline RL setup, which has not been considered in most of the fair RL literature.

The closest to our work is Mandal & Gan (2022) which adopted a welfare-based axiomatic approach and showed regret bound for Nash social welfare, and max-min fairness. In contrast, we considered the $\alpha$-based fairness metric and showed regret bound for the generic value of $\alpha$. Unlike in Mandal & Gan (2022), our approach admits efficient computation. Further, we provided the PAC guarantee and developed an algorithm for offline fair RL with a theoretical guarantee. Finally, we also developed a policy-gradient-based algorithm that is applicable to large state space as well unlike Mandal & Gan (2022).

**Convex and multi-objective MDP**: Our work is related to multi-objective RL (Roijers et al., 2013). Most of the approaches considered a single objective by weighing multiple objectives (Van Moffaert et al., 2013; Abels et al., 2019). Few also proposed algorithms to learn Pareto optimal front (Yang et al., 2019; Mossalam et al., 2016). We consider a non-linear function of the multiple value functions and provide a regret bound which was different from the existing approaches. Cheung (2019) obtained regret bound for a specific non-linear function of the objectives. Unlike the above, our objective is fairness among multiple agents. Naturally, the above papers did not consider the effect of various fairness metrics on the agents. Additionally, they considered infinite-horizon setup rather than an episodic setup which are fundamentally different. Thus, the algorithms are also different. Furthermore, we consider offline setup. Tarbouriech et al. (2020) considered a MDP setup with convex objective for infinite-horizon setup. However, the above paper also did not consider the fairness metrics and the offline which we considered. Furthermore, the paper considered sample complexity rather regret bound. Brantley et al. (2020) considered a MDP setting where the objective is to maximize a concave function of an individual value function. Unlike the above, we consider a setup where the objective is to achieve fairness among multiple agents. Naturally, the above paper did not consider the effect of various fairness metrics on the agents. Further, the proof techniques and algorithms also rely on the Lipschitz property, however, fairness function may not be Lipschitz (e.g.,max-min fairness), hence, the proof techniques are also different. Additionally, our proposed algorithm is computationally much simpler. For example, for the online setup, we only need to solve *a* convex optimization problem while the algorithm proposed by Brantley et al. (2020) is a bi-level optimization problem where the lower-level problems are convex. Hence, (Brantley et al., 2020)

needs to solve a large number of convex optimization problems at each episode. Furthermore, we consider offline setup unlike all the above papers.

**Multi-agent RL**: Another related approach is Multi-agent RL (MARL) which seeks to learn equilibrium (Li et al., 2022; Jin et al., 2021a) in the Markov game. However, our focus is to achieve fairness among the individual value functions. In our setup, the central controller is taking decisions rather than the agents. Hence, the objective is inherently different, and thus, the algorithms and analysis are also different.

However, in all the above setups, the fair policy is largely ignored. Besides, in the above papers, the individual agent takes a decision, instead, we focus on the setup where the central decision maker is taking the decision. Our focus is to ensure fairness among value functions of individual agents, hence, the setup is inherently different compared to the above.

## B  USEFUL LEMMAS

**Lemma 4.** *Let $x_1, x_2, \cdots, x_N$ and $y_1, y_2, \cdots, y_N$ be real numbers. We must have*

$$\left| \min_{i \in \{1,2,\cdots,N\}} x_i - \min_{j \in \{1,2,\cdots,N\}} y_j \right| \leq \max_{i \in [N]} |x_i - y_i|.$$

*Proof.* Without loss of generality, we let $\min_i x_i \geq \min_j y_j$. For any $i^* \in \arg\min_i x_i$ and any $j^* \in \arg\min_j y_j$, we have

$$x_{i^*} \leq x_{j^*},$$

which implies that

$$x_{i^*} - y_{j^*} \leq x_{j^*} - y_{j^*} \leq |x_{j^*} - y_{j^*}|. \tag{18}$$

We thus have

$$\left| \min_i x_i - \min_j y_j \right| = \min_i x_i - \min_j y_j$$
$$= x_{i^*} - y_{j^*}$$
$$\leq |x_{j^*} - y_{j^*}| \quad \text{(by Eq. (18))}$$
$$\leq \max_i |x_i - b_i|.$$

The result of this lemma thus follows. $\square$

**Lemma 5.** *Recall the definition of $C_F$ in Theorem 1. When $x_i, y_i \geq \epsilon > 0$ for all $i \in [N]$, we must have*

$$|F(x_1, x_2, \cdots, x_N) - F(y_1, y_2, \cdots, y_N)| \leq N \cdot C_F \max_{i \in [N]} |x_i - y_i|.$$

*Proof.* When $F = F_{\text{proportional}}$, we have

$$|F(x_1, x_2, \cdots, x_N) - F(y_1, y_2, \cdots, y_N)|$$
$$= \left| \sum_{i=1}^N \log x_i - \log y_i \right|$$
$$\leq \sum_{i=1}^N |\log x_i - \log y_i| \quad \text{(by the triangle inequality)}$$
$$\leq N \max_{i \in [N]} |\log x_i - \log y_i|$$
$$\leq N \frac{1}{\epsilon} \max_{i \in [N]} |x_i - y_i|.$$

The last step is by the Lipschitz continuity of $\log(\cdot)$ in the domain $[\epsilon, \infty)$, where $\frac{1}{\epsilon}$ is the corresponding Lipschitz constant.

Similarly, when $F = F_\alpha$, since the Lipschitz constant of $\frac{(\cdot)^{1-\alpha}}{1-\alpha}$ in the domain $[\epsilon, \infty)$ is $\epsilon^{-\alpha}$, we can show that

$$|F(x_1, x_2, \cdots, x_N) - F(y_1, y_2, \cdots, y_N)| \le N \cdot \epsilon^{-\alpha} \max_{i \in [N]} |x_i - y_i|.$$

When $F = F_{\text{max-min}}$, by Lemma 4, we have

$$|F(x_1, x_2, \cdots, x_N) - F(y_1, y_2, \cdots, y_N)| \le \max_{i \in [N]} |x_i - y_i|$$
$$= N \frac{1}{N} \cdot \max_{i \in [N]} |x_i - y_i|.$$

Notice that in this case $x_i, y_i \ge \epsilon$ is not needed.

In summary, the result of this lemma thus follows. □

**Lemma 6.** (10) *is a convex optimization whose value is monotone increasing w.r.t. the immediate reward $r$.*

*Proof.* Notice that the constraints of (10) are linear, we only need to prove that the fair objectives in Eqs. (4) to (6) are concave w.r.t. state-action occupancy measure $q$ and state-action-state occupancy measure $z$. Notice that $V^\pi_{1,(i)}$ is a weighted sum of $q$ and $z$, in order to prove the concavity, it remains to show that Eqs. (4) to (6) are concave w.r.t. $V^\pi_{1,(i)}$. Notice that $\min(\cdot)$ and $\log(\cdot)$ are concave. We only need to verify the concavity of $F_\alpha$. Since

$$\frac{\partial^2 \frac{1}{1-\alpha} x^{1-\alpha}}{\partial x^2} = -\alpha x^{-\alpha-1},$$

which is non-positive when $x \ge 0$. Thus, we have also proven the concavity of $F_\alpha$. Therefore, we have proven that (10) is a convex optimization.

Notice that $r$ only appears in the objective (i.e., the constraints do not have $r$), and all $F_{\text{max-min}}$, $F_{\text{proportional}}$, $F_\alpha$ are monotone increasing w.r.t. $r$. Thus, the value of (10) is monotone increasing w.r.t $r$.

The result of this lemma thus follows. □

**Lemma 7** (Hoeffding's inequality). *Let $Z_1, Z_2, \cdots, Z_n$ be i.i.d. samples of a random variable $Z \in [0, 1]$. For any $\tilde{\delta} > 0$, we must have*

$$\Pr\left\{ \left| \mathbb{E} Z - \frac{1}{n} \sum_{i=1}^n Z_i \right| \le \sqrt{\frac{\ln(2/\tilde{\delta})}{2n}} \right\} \ge 1 - \tilde{\delta}.$$

**Lemma 8** (empirical Bernstein inequality (Theorem 4 of (Maurer & Pontil, 2009))). *Let $Z_1, Z_2, \cdots, Z_n$ be i.i.d. samples of a random variable $Z \in [0, 1]$. For any $\tilde{\delta} > 0$, we must have*

$$\Pr\left\{ \left| \mathbb{E} Z - \frac{1}{n} \sum_{i=1}^n Z_i \right| \le \sqrt{\frac{2V_n \ln(4/\tilde{\delta})}{n}} + \frac{7\ln(4/\tilde{\delta})}{3(n-1)} \right\} \ge 1 - \tilde{\delta},$$

*where $\text{VAR}_n$ is the sample variance*

$$\text{VAR}_n = \frac{1}{n(n-1)} \sum_{1 \le i \le j \le n} (Z_i - Z_j)^2. \tag{19}$$

**Lemma 9** (Lemma F.4 of (Dann et al., 2017)). *Let $\mathcal{F}_i$ for $i = 1, 2, \cdots$ be a filtration and $X_1, X_2, \cdots$ be a sequence of Bernoulli random variables with $\Pr\{X_i = 1 | \mathcal{F}_{i-1}\} = P_i$ with $P_i$ being $\mathcal{F}_{i-1}$-measurable and $X_i$ being $F_i$ measurable. For any $W \ge 0$, It holds that*

$$\Pr\left\{ exist\ n : \sum_{t=1}^n X_t < \sum_{t=1}^n \frac{P_t}{2} - W \right\} \le e^{-W}.$$

The following is the standard value difference lemma. Its proof can be found in, e.g., (Dann et al., 2017), Lemma E.15.

**Lemma 10** (Value difference lemma). *For any two MDPs $M'$ and $M''$ with rewards $r'$ and $r''$ and transition probabilities $P'$ and $P''$, the difference in value functions with respect to the same policy $\pi$ can be written as*

$$V_i'(s) - V_i''(s) = \mathop{\mathbb{E}}_{P'',\pi} \left[ \sum_{t=i}^{H} (r'(s_t, a_t, t) - r''(s_t, a_t, t)) \,\middle|\, s_i = s \right]$$
$$+ \mathop{\mathbb{E}}_{P'',\pi} \left[ \sum_{t=i}^{H} \sum_{\tilde{s}} \left( P_t'(\tilde{s}|s_t, a_t) - P_t''(\tilde{s}|s_t, a_t) \right)^T V_{t+1}'(\tilde{s}) \right].$$

## C   DETAILS IN SECTION 5

The overall confidence interval is defined as

$$M_k := \{ (\tilde{p}, \tilde{r}) : \text{Eq. (11) and Eq. (12)} \},$$

i.e., the true value of $(p, r)$ is in $M_k$ with high probability. We now want to use this confidence interval $M_k$ in (10). A possible way is to replace $p$ and $r$ by $(\tilde{r}, \tilde{p}) \in M_k$ and view $\tilde{r}, \tilde{p}$ as decision variables. Thus, the objective in (10) now becomes

$$\max_{(\tilde{r}, \tilde{p}) \in M_k, \ q \in \mathcal{Q}} V_1^{\pi, F}(s_1). \tag{20}$$

It is indeed an optimistic solution compared to the real optimal solution because we relax the value of $r$ and $p$ in such optimization (which leads to better objective value). However, it is no longer a convex optimization problem because now $r$ and $p$ are decision variables. To turn such an optimization into a convex one, we need to determine the value of $\tilde{r}$ and $\tilde{p}$ beforehand. To that end, notice the monotonicity of the objective with respect to $\tilde{r}$ (proof in Lemma 6 in Appendix B). Thus, without affecting the solution of (20), we can let

$$\tilde{r}_{h,(i)}(s, a) = \overline{r}_{h,(i)}^{k-1}(s, a) + \beta_{h,k}^r(s, a). \tag{21}$$

Now, we only need to determine the value of $\tilde{p}$. To that end, consider the state-action-next-state occupancy measure $z_h^\pi(s, a, s'; p) := p_h(s'|s, a) q_h^\pi(s, a; p)$. Considering Eq. (11), we only need

$$z_h(s, a, s') \le \left( \overline{p}_h^{k-1}(s'|s, a) + \beta_{h,k}^p(s, a, s') \right) \sum_{y \in \mathcal{S}} z_h(s, a, y) \text{ for all } h \in [H-1], s, a, s',$$
$$z_h(s, a, s') \ge \left( \overline{p}_h^{k-1}(s'|s, a) - \beta_{h,k}^p(s, a, s') \right) \sum_{y \in \mathcal{S}} z_h(s, a, y) \text{ for all } h \in [H-1], s, a, s'. \tag{22}$$

Now we are ready to solve (20) by convex optimization.

### C.1   ABOUT OPTIMIZATION PROBLEMS

In this subsection, we will show details of the optimization problems (10), (20), and (13).

Let $\mu(s)$ denote the probability of the initial state $s$. (for a fixed initial state $s_1$, then $\mu(s)$ equals to 1 for $s = s_1$ while equals to 0 otherwise.)

**About $\mathcal{Q}$ (constraints of $q$):**

The following are the constraints that make $q$ a legit state-action occupancy measure, i.e., the definition of $\mathcal{Q}$:

$$\begin{aligned}
\sum_a q_h(s, a) &= \sum_{s', a'} p_{h-1}(s|s', a') q_{h-1}(s', a') &&\text{for all } s \in \mathcal{S}, h \in [H] \setminus \{1\} \\
q_h(s, a) &\ge 0 &&\text{for all } s \in \mathcal{S}, a \in \mathcal{A}, h \in [H] \\
\sum_a q_1(s, a) &= \mu(s) &&\text{for all } s \in \mathcal{S}.
\end{aligned} \tag{23}$$

The constraint $\sum_{s,a} q_h(s,a) = 1$ is redundant because the first and the third constraint imply $\sum_{s,a} q_h(s,a) = 1$ for all $h \in [H]$.

**About $\mathcal{Z}$ (constraints of $z$):**

By the definition of $z$, (22) is part of $\mathcal{Z}$. Besides, $z$ should also be constrained by Eq. (23). Recall the definition of $z$

$$z_h(s,a,s') := p_h(s'|s,a)q_h(s,a). \tag{24}$$

By summing over the next state $s'$ on both sides of Eq. (24), we have

$$\sum_{s'} z_h(s,a,s') = \sum_{s'} p_h(s'|s,a)q_h(s,a) = q_h(s,a). \tag{25}$$

Summing over $a$ on both sides of Eq. (25), we have

$$\sum_{a,s'} z_h(s,a,s') = \sum_a q_h(s,a). \tag{26}$$

Thus, we can rewrite the constraints $\mathcal{Q}$ Eq. (23) in the form of $z$:

$$
\begin{aligned}
\sum_{a,s'} z_h(s,a,s') &= \sum_{s',a'} z_{h-1}(s',a',s) && \text{for all } s \in \mathcal{S}, h \in [H] \setminus \{1\}, \\
z_h(s,a,s') &\geq 0 && \text{for all } s,a,s',h, \\
\sum_{a,s'} z_1(s,a,s') &= \mu(s) && \text{for all } s \in \mathcal{S}.
\end{aligned}
\tag{27}
$$

In Eq. (27), we get the first constraint by plugging Eq. (26) into the left side of the first constraint of Eq. (23) while plugging Eq. (24) into the right side. We get the third constraint by plugging Eq. (26) into the third constraint of Eq. (23).

Notice that by replacing $q$ by $z$, we have one additional requirement Eq. (24). Using Eq. (25) to replace $q_h(s,a)$ in Eq. (24), we can express Eq. (24) as

$$z_h(s,a,s') = p_h(s'|s,a) \cdot \sum_{y \in \mathcal{S}} z_h(s,a,y). \tag{28}$$

By Eqs. (11) and (28), we have the constraint Eq. (22) (used in the optimization problem (13)).

We now show that (20) is equivalent to (13) by the following proposition.

**Proposition 11.** (20) *and* (13) *are equivalent. In other words, the optimal value of the objective of* (20) *is equal to the optimal value of the objective of* (13).

*Proof.* By the monotonicity w.r.t. $r$ shown in Lemma 6, we know that the optimal choice of $\tilde{r}$ in (20) is $\bar{r} + \beta^r$. It remains to show that the effect of choosing optimal of $\tilde{p}$ in (20) is equivalent to Eq. (22). To that end, notice that $\tilde{p}$ does not appear in the objective. Thus, we only need to focus on how $\tilde{p}$ affects the constraints of $z$. Notice that among all constraints in Eqs. (27) and (28), the only one that connects $p$ and $z$ is Eq. (28). Since the optimal $\tilde{p}$ in (20) must be in the confidence interval $[\bar{p} - \beta^p, \bar{p} + \beta^p]$, we know that the optimal objective value by using Eq. (22) is at least as good as the one by using the optimal $\tilde{p}$ in Eq. (28). From another aspect, For the optimal $z$ get by Eq. (22), we can always construct $\tilde{p}$ which is in the confidence interval $[\bar{p} - \beta^p, \bar{p} + \beta^p]$ by letting

$$
\tilde{p}_h(s'|s,a) = \begin{cases} \frac{z_h(s,a,s')}{\sum_{y \in \mathcal{S}} z_h(s,a,y)} & \text{if } \sum_{y \in \mathcal{S}} z_h(s,a,y) \neq 0, \\ \bar{p} & \text{if } \sum_{y \in \mathcal{S}} z_h(s,a,y) = 0. \end{cases}
$$

which implies that the optimal objective value by using optimal $\tilde{p}$ in Eq. (28) is at least as good as the one by using Eq. (22). The equivalent of these two different approaches is thus follows. $\qquad\square$

### C.2 PROOF OF THEOREM 1

To prove Theorem 1, we will first introduce the good event and its probability, then prove a regret bound under the good event. Some auxiliary lemmas are needed in the proof. We list them at the end of this subsection.

**Failure events and the good event**

Define the empirical average of the transition probability and the immediate reward at the $k$-th iteration of Algorithm 1 as

$$n_h^{k-1}(s,a) := \sum_{k'=1}^{k-1} \mathbb{1}\left(s_h^{k'} = s, a_h^{k'} = a\right), \tag{29}$$

$$\overline{p}_h^{k-1}(s'|s,a) := \frac{\sum_{k'=1}^{k-1} \mathbb{1}\left(s_h^{k'} = s, a_h^{k'} = a, s_{h+1}^{k'} = s'\right)}{\max\{n_h^{k-1}(s,a),\, 1\}}, \tag{30}$$

$$\overline{r}_{h,(i)}^{k-1}(s,a) := \frac{\sum_{k'=1}^{k-1} \hat{r}_{h,(i)}^{k'}(s,a) \cdot \mathbb{1}\left(s_h^{k'} = s, a_h^{k'} = a\right)}{\max\{n_h^{k-1}(s,a),\, 1\}}. \tag{31}$$

Define

$$\beta_{h,k}^p(s,a,s') := \sqrt{\frac{4\overline{p}_h^{k-1}(s'|s,a)(1 - \overline{p}_h^{k-1}(s'|s,a))L_\delta^p}{\max\{n_h^{k-1}(s,a),1\}}} + \frac{14L_\delta^p}{3\max\{n_h^{k-1}(s,a),1\}}, \tag{32}$$

$$\beta_{h,k}^r(s,a) := \sqrt{\frac{L_\delta^r}{\max\{n_h^{k-1}(s,a),1\}}}. \tag{33}$$

where $L_\delta^p := \ln \frac{12S^2 AHK}{\delta}$ and $L_\delta^r := 2\ln \frac{3SAHNK}{\delta}$.

We define the following failure events based on confidence intervals in Eq. (11) and Eq. (12).

$$G^p := \left\{ \text{exist some } s,a,s',h,k \text{ such that } \left|\overline{p}_h^{k-1}(s'|s,a) - p_h(s'|s,a)\right| \geq \beta_{h,k}^p(s,a,s') \right\},$$

$$G^n := \left\{ \text{exist some } s,a,h,k \text{ such that } n_h^{k-1}(s,a) \leq \frac{1}{2}\sum_{j<k} q_h^{\pi_j}(s,a) - \ln \frac{3SAH}{\delta} \right\}, \tag{34}$$

$$G^r := \left\{ \text{exist some } s,a,h,i,k \text{ such that } \left|\overline{r}_{h,(i)}^{k-1}(s,a) - r_{h,(i)}^{k-1}(s,a)\right| \geq \beta_{h,k}^r(s,a) \right\}.$$

Intuitively, $G^p$ denotes the case where the transition probability is out of the confidence interval, $G^n$ denotes the case where the empirical occupancy measure deviates from the actual occupancy measure, and $G^r$ denotes the case where the empirical reward is out of the confidence interval. The following lemma estimates the probability of those failure events.

**Lemma 12.** *We have*

$$\Pr\{G^p\} \leq \frac{\delta}{3}.$$

*Proof.* We first focus on the situation on fixed $s,a,s',h,k$. If $n_h^{k-1}(s,a) \in \{0,1\}$, then

$$\beta_{h,k}^p(s,a,s') = \sqrt{4\overline{p}_h^{k-1}(s'|s,a)(1 - \overline{p}_h^{k-1}(s'|s,a))L_\delta^p} + \frac{14L_\delta^p}{3} \geq \frac{14L_\delta^p}{3} \geq \frac{14\ln 4}{3} > 2.$$

Thus, we have

$$\Pr\left\{\left|\overline{p}_h^{k-1}(s'|s,a) - p_h(s'|s,a)\right| \geq \beta_{h,k}^p(s,a,s') \mid n_h^{k-1}(s,a) \in \{0,1\}\right\} = 0 \leq \frac{\delta}{3S^2 AHK}. \tag{35}$$

Now we consider the case of $n_h^{k-1}(s,a) \geq 2$. We define $b_1, b_2, \cdots, b_{n_h^{k-1}(s,a)}$, where each term is $\mathbb{1}\left(s_{h+1}^{k'} = s'\right)$ under the condition $s_h^{k'} = s$ and $a_h^{k'} = a$ for $k = 1, 2, \cdots, k-1$ (recall the definition of $n_h^{k-1}(s,a)$ in Eq. (29)). Thus, $b_1, b_2, \cdots, b_{n_h^{k-1}(s,a)}$ are $n_h^{k-1}(s,a)$ i.i.d. samples of Bernoulli distribution with the parameter of the (success) probability $p_h(s'|s,a)$. Therefore, the sample variance (defined in Eq. (19)) of these $n_h^{k-1}(s,a)$ samples is equal to

$$\frac{1}{n_h^{k-1}(s,a)(n_h^{k-1}(s,a)-1)} \sum_{1 \leq i \leq j \leq n_h^{k-1}(s,a)-1} (b_i - b_j)^2$$

$$= \frac{\sum_{k'=1}^{k-1} \mathbb{1}\left(s_h^{k'} = s, a_h^{k'} = a, s_{h+1}^{k'} = s'\right) \cdot \sum_{k'=1}^{k-1} \mathbb{1}\left(s_h^{k'} = s, a_h^{k'} = a, s_{h+1}^{k'} \neq s'\right)}{n_h^{k-1}(s,a)(n_h^{k-1}(s,a)-1)}$$

$$= \frac{n_h^{k-1}(s,a)}{n_h^{k-1}(s,a)-1} \overline{p}_h^{k-1}(s'|s,a)(1 - \overline{p}_h^{k-1}(s'|s,a)) \quad \text{(by Eq. (30))}$$

$$\leq 2\overline{p}_h^{k-1}(s'|s,a)(1 - \overline{p}_h^{k-1}(s'|s,a)).$$

Thus, by Lemma 8 (where $\tilde{\delta} = \frac{\delta}{3S^2 AHK}$), for fixed $s, a, s', h, k$, we have

$$\Pr\left\{ \left| \overline{p}_h^{k-1}(s'|s,a) - p_h(s'|s,a) \right| \geq \sqrt{\frac{4\overline{p}_h^{k-1}(s'|s,a)(1 - \overline{p}_h^{k-1}(s'|s,a)) \ln \frac{12S^2 AHK}{\delta}}{n_h^{k-1}(s,a)}} \right.$$

$$\left. + \frac{7 \ln \frac{12S^2 AHK}{\delta}}{3(n_h^{k-1}(s,a)-1)} \right\} \leq \frac{\delta}{3S^2 AHK}.$$

Notice that $\frac{7}{3(n_h^{k-1}(s,a)-1)} \leq \frac{14}{3}$ when $n_h^{k-1}(s,a) \geq 2$. We thus have

$$\Pr\left\{ \left| \overline{p}_h^{k-1}(s'|s,a) - p_h(s'|s,a) \right| \geq \beta_{h,k}^p(s,a,s') \mid n_h^{k-1}(s,a) \geq 2 \right\} \leq \frac{\delta}{3S^2 AHK}. \tag{36}$$

Combining Eq. (35) and Eq. (36), we thus have

$$\Pr\left\{ \left| \overline{p}_h^{k-1}(s'|s,a) - p_h(s'|s,a) \right| \geq \beta_{h,k}^p(s,a,s') \right\} \leq \frac{\delta}{3S^2 AHK}.$$

Applying the union bound by traversing all $s, a, s', h, k$, we thus have

$$\Pr\{G^p\} \leq \frac{\delta}{3}.$$

The result of this lemma thus follows. □

**Lemma 13.** *We have*

$$\Pr\{G^n\} \leq \frac{\delta}{3}.$$

*Proof.* For fixed $s, a, h$, by Lemma 9 (letting $W = \ln \frac{3SAH}{\delta}$), we have

$$\Pr\left\{ \text{exist } k \text{ such that } n_h^{k-1}(s,a) \leq \frac{1}{2} \sum_{j<k} q_h^{\pi_j}(s,a|p) - \ln \frac{3SAH}{\delta} \right\} \leq \frac{\delta}{3SAH}.$$

Applying the union bound by traversing all $s, a, h$, the result of this lemma thus follows. □

**Lemma 14.** *We have*

$$\Pr\{G^r\} \leq \frac{\delta}{3}.$$

*Proof.* For fixed $s, a, h, i, k$, by Lemma 7, we have

$$\Pr\left\{\left|\bar{r}_{h,(i)}^{k-1}(s,a) - r_{h,(i)}^{k-1}(s,a)\right| \geq \sqrt{\frac{L_\delta^r}{n_h^{k-1}(s,a)}}\right\} \leq \frac{\delta}{3SAHNK}$$

Applying the union bound by traversing all $s, a, h, i, k$, the result of this lemma thus follows. □

**The regret bound under the good event**

**Lemma 15.** *If outside the union of all failure events $G^p \cup G^n \cup G^r$, then we must have*

$$\begin{aligned}
\text{Reg}(K) \leq & 4C_F\sqrt{L_\delta^r \ln(4+K)}HN\sqrt{SAK} + 2C_F\sqrt{L_\delta^r}\left(4\ln\frac{SAH}{\delta'} + 5\right)HNSA \\
& + 8C_F\sqrt{L_\delta^p \ln(4+K)}H^2NS\sqrt{AK} + 4C_F\sqrt{L_\delta^p}\left(4\ln\frac{SAH}{\delta'} + 5\right)HN^2S^{3/2}A \\
& + \frac{28C_FL_\delta^p\left(4\ln(4+K) + 4\ln\frac{SAH}{\delta'} + 5\right)}{3}H^2NS^2A \\
= & C_F \cdot \left(\tilde{\mathcal{O}}(H^2NS\sqrt{AK}) + \tilde{\mathcal{O}}(HN^2S^{3/2}A) + \tilde{\mathcal{O}}(H^2NS^2A)\right).
\end{aligned}$$

*Proof.* Let $\tilde{r}^k, \tilde{p}^k$, and $\pi^k$ denote the optimal $\tilde{r}, \tilde{p}$, and policy in (20) in the $k$-th iteration of Algorithm 1, respectively. Since outside $G^r$, we have $\tilde{r}^k \geq r$. Thus, by Assumption 1 and Remark 1, we have

$$V_{1,(i)}^{\pi_k}(s_1; \tilde{r}^k, \tilde{p}^k) \geq \epsilon, \quad \text{and} \quad V_{1,(i)}^{\pi_k}(s_1; r, p) \geq \epsilon. \tag{37}$$

We have

$$\begin{aligned}
& \text{Reg}(K) \\
= & \sum_{k=1}^K \left(V_1^*(s_1) - V_1^{\pi_k}(s_1)\right) \\
= & \sum_{k=1}^K \left(V_1^*(s_1; r, p) - V_1^{\pi_k}(s_1; r, p)\right) \\
\leq & \sum_{k=1}^K \left(V_1^{\pi_k}(s_1; \tilde{r}^k, \tilde{p}^k) - V_1^{\pi_k}(s_1; r, p)\right) \text{ (by optimism, i.e., } \tilde{r}^k \text{ and } \tilde{q}^k \text{ are optimal)} \\
\leq & \sum_{k=1}^K C_F N \max_{i \in [N]}\left|V_{1,(i)}^{\pi_k}(s_1; \tilde{r}^k, \tilde{p}^k) - V_{1,(i)}^{\pi_k}(s_1; r, p)\right| \text{ (by Lemma 5 and Eq. (37))} \\
= & C_F N \sum_{k=1}^K \max_{i \in [N]}\left|\mathbb{E}\sum_{h=1}^H\left[\left(\tilde{r}_{h,(i)}^{k-1}(s,a) - r_{h,(i)}(s_h, a_h)\right)\right.\right. \\
& + \left.\left.\sum_{s' \in \mathcal{S}}\left(\tilde{p}_h^{k-1}(s'|s,a) - p_h(s'|s,a)\right) \cdot V_{h+1,(i)}^{\pi_k}(s'; \tilde{r}_{(i)}^{k-1}, \tilde{p})\right|s_1, p, \pi_k\right]\right| \text{ (by Lemma 10)} \\
\leq & \underbrace{C_F N \sum_{k=1}^K \max_{i \in [N]}\mathbb{E}\left[\sum_{h=1}^H\left|\tilde{r}_{h,(i)}^{k-1}(s,a) - r_{h,(i)}(s_h, a_h)\right| \middle| s_1, p, \pi_k\right]}_{\text{Term A}} \\
& + \underbrace{C_F N \sum_{k=1}^K \max_{i \in [N]}\mathbb{E}\left[\sum_{h=1}^H\sum_{s'}\left|\tilde{p}_h^{k-1}(s'|s,a) - p_h(s'|s,a)\right| \cdot \left|V_{h+1,(i)}^{\pi_k}(s'; \tilde{r}_{(i)}^{k-1}, \tilde{p})\right| \middle| s_1, p, \pi_k\right]}_{\text{Term B}}.
\end{aligned}$$

$$\tag{38}$$

Since outside $G^r$ and $G^p$, we have

$$\left|\tilde{r}_{h,(i)}^{k-1}(s,a) - r_{h,(i)}(s_h,a_h)\right| \le 2\beta_{h,k}^r(s,a), \tag{39}$$

$$\left|\tilde{p}_h^{k-1}(s'|s,a) - p_h(s'|s,a)\right| \le 2\beta_{h,k}^p(s,a,s'). \tag{40}$$

We have

Term A of Eq. (38)

$$\le 2C_F N \sum_{k=1}^{K}\sum_{h=1}^{H}\sum_{(s,a)} q_h^{\pi_k}(s,a)\beta_{h,k}^r(s,a) \quad \text{(by Eq. (39))}$$

$$= 2C_F N \sum_{k=1}^{K}\sum_{h=1}^{H}\sum_{(s,a)} q_h^{\pi_k}(s,a)\sqrt{\frac{L_\delta^r}{\max\{n_h^{k-1}(s,a),1\}}} \quad \text{(by Eq. (33))}$$

$$\le 2C_F \sqrt{L_\delta^r} N \left(2H\sqrt{SAK\ln(4+K)} + SAH\left(4\ln\frac{SAH}{\delta'}+5\right)\right) \quad \text{(by Lemma 16)}$$

$$= 4C_F \sqrt{L_\delta^r \ln(4+K)} HN\sqrt{SAK} + 2C_F\sqrt{L_\delta^r}\left(4\ln\frac{SAH}{\delta'}+5\right)HNSA.$$

Since $0 \le \tilde{r}_{h,(i)}^{k-1}(s,a) \le 1$ for all $h,i,s,a$, we have

$$\left|V_{h+1,(i)}^{\pi_k}(s';\tilde{r}_{(i)}^{k-1},\tilde{p})\right| \le H. \tag{41}$$

Thus, we have

Term B of Eq. (38)

$$\le 2C_F NH \sum_{k=1}^{K}\sum_{h=1}^{H}\sum_{(s,a)} q_h^{\pi_k}(s,a)\sum_{s'}\beta_{h,k}^p(s,a,s') \quad \text{(by Eqs. (40) and (41))}$$

$$\le 4C_F HN\sqrt{L_\delta^p}\sum_{k=1}^{K}\sum_{h=1}^{H}\sum_{(s,a)} q_h^{\pi_k}(s,a)\frac{1}{\sqrt{\max\{n_h^{k-1}(s,a),1\}}}\sum_{s'}\sqrt{\overline{p}_h^{k-1}(s'|s,a)}$$

$$+ \frac{28C_F HNSL_\delta^p}{3}\sum_{k=1}^{K}\sum_{h=1}^{H}\sum_{(s,a)}\frac{q_h^{\pi_k}(s,a)}{\max\{n_h^{k-1}(s,a),1\}} \quad \text{(by Eq. (32) and } 1-\overline{p}_h^{k-1}(s'|s,a)\le 1)$$

$$\le 4C_F HN\sqrt{L_\delta^p}\sum_{k=1}^{K}\sum_{h=1}^{H}\sum_{(s,a)} q_h^{\pi_k}(s,a)\frac{\sqrt{S}}{\sqrt{\max\{n_h^{k-1}(s,a),1\}}}\sqrt{\sum_{s'}\overline{p}_h^{k-1}(s'|s,a)}$$

$$+ \frac{28C_F HNSL_\delta^p}{3}\sum_{k=1}^{K}\sum_{h=1}^{H}\sum_{(s,a)}\frac{q_h^{\pi_k}(s,a)}{\max\{n_h^{k-1}(s,a),1\}} \quad \text{(by Cauchy–Schwarz inequality)}$$

$$= 4C_F HN\sqrt{SL_\delta^p}\sum_{k=1}^{K}\sum_{h=1}^{H}\sum_{(s,a)} q_h^{\pi_k}(s,a)\frac{1}{\sqrt{\max\{n_h^{k-1}(s,a),1\}}}$$

$$+ \frac{28C_F HNSL_\delta^p}{3}\sum_{k=1}^{K}\sum_{h=1}^{H}\sum_{(s,a)}\frac{q_h^{\pi_k}(s,a)}{\max\{n_h^{k-1}(s,a),1\}} \quad \text{(since } \sum_{s'}\overline{p}_h^{k-1}(s'|s,a)=1)$$

$$\le 8C_F\sqrt{L_\delta^p \ln(4+K)}H^2 NS\sqrt{AK} + 4C_F\sqrt{L_\delta^p}\left(4\ln\frac{SAH}{\delta'}+5\right)HN^2 S^{3/2}A$$

$$+ \frac{28C_F L_\delta^p\left(4\ln(4+K)+4\ln\frac{SAH}{\delta'}+5\right)}{3}H^2 NS^2 A \quad \text{(by Lemma 16)}.$$

$\square$

**Some auxiliary lemmas**

Define $\delta' := \frac{\delta}{3}$ and

$$L_{k,h} := \left\{ (s,a) \; \middle| \; \frac{1}{4} \sum_{j<k} q_h^{\pi_j}(s,a) \geq \ln \frac{SAH}{\delta'} + 1 \right\}. \tag{42}$$

The following lemmas and proofs are similar to those in (Efroni et al., 2019; Zanette & Brunskill, 2019) with different notations. For ease of reading, we provide the full proof using the notation of this paper.

**Lemma 16.** *If outside the failure event $G^n$, then*

$$\sum_{k=1}^{K} \sum_{h=1}^{H} \sum_{s,a} q_h^{\pi_k}(s,a) \sqrt{\frac{1}{\max\{n_h^{k-1}(s,a),1\}}} \leq 2H\sqrt{SAK\ln(4+K)} + SAH\left(4\ln\frac{SAH}{\delta'} + 5\right),$$
$$\tag{43}$$

$$\sum_{k=1}^{K} \sum_{h=1}^{H} \sum_{s,a} \frac{q_h^{\pi_k}(s,a)}{\max\{n_h^{k-1}(s,a),1\}} \leq SAH\left(4\ln(4+K) + 4\ln\frac{SAH}{\delta'} + 5\right). \tag{44}$$

*Proof.* We have

$$\sum_{k=1}^{K} \sum_{h=1}^{H} \sum_{s,a} q_h^{\pi_k}(s,a) \sqrt{\frac{1}{\max\{n_h^{k-1}(s,a),1\}}}$$

$$\leq \sum_{k=1}^{K} \sum_{h=1}^{H} \sum_{s,a} q_h^{\pi_k}(s,a) \sqrt{\frac{1}{n_h^{k-1}(s,a)}} \quad \text{(since } n_h^{k-1}(s,a) \geq 1 \text{ by Lemma 17)}$$

$$\leq \sum_{k=1}^{K} \sum_{h=1}^{H} \sum_{(s,a)\in L_{k,h}} q_h^{\pi_k}(s,a) \sqrt{\frac{1}{n_h^{k-1}(s,a)}} + \sum_{k=1}^{K} \sum_{h=1}^{H} \sum_{(s,a)\notin L_{k,h}} q_h^{\pi_k}(s,a). \tag{45}$$

By Eq. (8), we have

$$\sum_{k=1}^{K} \sum_{h=1}^{H} \sum_{(s,a)\in L_{k,h}} q_h^{\pi_k}(s,a) \leq \sum_{k=1}^{K} \sum_{h=1}^{H} \sum_{(s,a)} q_h^{\pi_k}(s,a) = KH. \tag{46}$$

For the first term of Eq. (45), we have

$$\sum_{k=1}^{K} \sum_{h=1}^{H} \sum_{(s,a)\in L_{k,h}} q_h^{\pi_k}(s,a) \sqrt{\frac{1}{n_h^{k-1}(s,a)}}$$

$$= \sum_{k=1}^{K} \sum_{h=1}^{H} \sum_{(s,a)\in L_{k,h}} \sqrt{q_h^{\pi_k}(s,a)} \sqrt{\frac{q_h^{\pi_k}(s,a)}{n_h^{k-1}(s,a)}}$$

$$\leq \sqrt{\sum_{k=1}^{K} \sum_{h=1}^{H} \sum_{(s,a)\in L_{k,h}} q_h^{\pi_k}(s,a)} \cdot \sqrt{\sum_{k=1}^{K} \sum_{h=1}^{H} \sum_{(s,a)\in L_{k,h}} \frac{q_h^{\pi_k}(s,a)}{n_h^{k-1}(s,a)}} \quad \text{(by Cauchy-Schwarz inequality)}$$

$$\leq \sqrt{KH}\sqrt{4SAH\ln(4+K)} \quad \text{(by Eq. (46) and Lemma 20)}$$

$$= 2H\sqrt{SAK\ln(4+K)}. \tag{47}$$

By Eq. (47), Eq. (45), and Lemma 19, we can get Eq. (43). It remains to prove Eq. (44). To that end, we have

$$\sum_{k=1}^{K}\sum_{h=1}^{H}\sum_{s,a}\frac{q_h^{\pi_k}(s,a)}{\max\{n_h^{k-1}(s,a),1\}}$$

$$\leq\sum_{k=1}^{K}\sum_{h=1}^{H}\sum_{s,a}\frac{q_h^{\pi_k}(s,a)}{n_h^{k-1}(s,a)}\quad\text{(since }n_h^{k-1}(s,a)\geq 1\text{ by Lemma 17)}$$

$$\leq\sum_{k=1}^{K}\sum_{h=1}^{H}\sum_{(s,a)\in L_{k,h}}\frac{q_h^{\pi_k}(s,a)}{n_h^{k-1}(s,a)}+\sum_{k=1}^{K}\sum_{h=1}^{H}\sum_{(s,a)\notin L_{k,h}}q_h^{\pi_k}(s,a)$$

$$\leq 4SAH\ln(4+K)+SAH\left(4\ln\frac{SAH}{\delta'}+5\right)\quad\text{(by Lemma 19 and Lemma 20)}$$

$$=SAH\left(4\ln(4+K)+4\ln\frac{SAH}{\delta'}+5\right).$$

Thus, we have proven Eq. (44). The result of this lemma thus follows. $\qquad\square$

**Lemma 17.** *If outside the failure event $G^n$, for any $(s,a)\in L_{k,h}$, we must have*

$$n_h^{k-1}(s,a)\geq\max\left\{\frac{1}{4}\sum_{j\leq k}q_h^{\pi_j}(s,a),\ 1\right\}.$$

*Proof.* We have

$$n_h^{k-1}(s,a)>\frac{1}{4}\sum_{j<k}q_h^{\pi_j}(s,a)+\frac{1}{4}\sum_{j<k}q_h^{\pi_j}(s,a)-\ln\frac{SAH}{\delta'}\quad\text{(recall the definition of }G^n\text{ in Eq. (34))}$$

$$\geq\frac{1}{4}\sum_{j<k}q_h^{\pi_j}(s,a)+1\quad\text{(since }(s,a)\in L_{k,h})$$

$$\geq\max\left\{\frac{1}{4}\sum_{j\leq k}q_h^{\pi_j}(s,a),\ 1\right\}\quad\text{(since }q_h^{\pi_k}\leq 1).$$

$\qquad\square$

**Lemma 18.** *For any $(s,a)\notin L_{k,h}$, we must have*

$$\sum_{j\leq k}q_h^{\pi_j}(s,a)\leq 4\ln\frac{SAH}{\delta'}+5.$$

*Proof.* Since $(s,a)\notin L_{k,h}$, we have

$$\frac{1}{4}\sum_{j<k}q_h^{\pi_j}(s,a)<\ln\frac{SAH}{\delta'}+1.$$

Thus, we have

$$\sum_{j<k}q_h^{\pi_j}(s,a)<4\ln\frac{SAH}{\delta'}+4.$$

Because $q_h^{\pi_k}\leq 1$, the result of this lemma thus follows. $\qquad\square$

**Lemma 19.** *We have*

$$\sum_{k=1}^{K}\sum_{h=1}^{H}\sum_{(s,a)\notin L_{k,h}}q_h^{\pi_k}(s,a)\leq SAH\left(4\ln\frac{SAH}{\delta'}+5\right).$$

*Proof.* Define

$$k_{s,a,h} := \begin{cases} 0, & \text{if } \{k^* \in [K] \mid (s,a) \notin L_{k^*,h}\} = \varnothing, \\ \max\{k^* \in [K] \mid (s,a) \notin L_{k^*,h}\} & \text{otherwise.} \end{cases} \quad (48)$$

By the definition of $L_{k,h}$ in Eq. (42), we know that

$$(s,a) \notin L_{k,h} \text{ for all } k \leq k_{s,a,h}. \quad (49)$$

Therefore, we have

$$\sum_{k=1}^{K}\sum_{h=1}^{H}\sum_{(s,a)\notin L_{k,h}} q_h^{\pi_k}(s,a) = \sum_{(s,a)}\sum_{k=1}^{K}\sum_{h=1}^{H} q_h^{\pi_k}(s,a)\mathbb{1}\left((s,a)\notin L_{k,h}\right)$$

$$= \sum_{(s,a)}\sum_{h=1}^{H}\sum_{k=1}^{k_{s,a,h}} q_h^{\pi_k}(s,a) \text{ (by Eq. (48))}$$

$$\leq SAH\left(4\ln\frac{SAH}{\delta'}+5\right) \text{ (by Eq. (49) and Lemma 18).}$$

$\square$

**Lemma 20.** *If outside the failure event $G^n$, we must have*

$$\sum_{k=1}^{K}\sum_{h=1}^{H}\sum_{(s,a)\in L_{k,h}} \frac{q_h^{\pi_k}(s,a)}{n_h^{k-1}(s,a)} \leq 4SAH\ln(4+K).$$

*Proof.* We have

$$\sum_{k=1}^{K}\sum_{h=1}^{H}\sum_{(s,a)\in L_{k,h}} \frac{q_h^{\pi_k}(s,a)}{n_h^{k-1}(s,a)} \leq 4\sum_{k=1}^{K}\sum_{h=1}^{H}\sum_{(s,a)\in L_{k,h}} \frac{q_h^{\pi_k}(s,a)}{\sum_{j\leq k} q_h^{\pi_j}(s,a)} \text{ (by Lemma 17)}$$

$$= 4\sum_{(s,a)}\sum_{h=1}^{H}\sum_{k=1}^{K} \frac{q_h^{\pi_k}(s,a)}{\sum_{j\leq k} q_h^{\pi_j}(s,a)}\mathbb{1}\left((s,a)\in L_{k,h}\right). \quad (50)$$

For fixed $s,a,h$, if

$$\{k=1,2,\cdots,\mid(s,a)\in L_{k,h}\} \neq \varnothing,$$

then by the monotonicity of the size of $L_{k,h}$ with respect to $k$, we can define

$$k_{s,a,h} := \min\{k=1,2,\cdots,\mid(s,a)\in L_{k,h}\}.$$

Thus, we have

$$\sum_{k=1}^{K} \frac{q_h^{\pi_k}(s,a)}{\sum_{j\leq k} q_h^{\pi_j}(s,a)}\mathbb{1}\left((s,a)\in L_{k,h}\right) = \sum_{k=k_{s,a,h}}^{K} \frac{q_h^{\pi_k}(s,a)}{\sum_{j\leq k} q_h^{\pi_j}(s,a)}$$

$$\leq \sum_{k=k_{s,a,h}}^{K} \frac{q_h^{\pi_k}(s,a)}{4 + \sum_{k_{s,a,h}\leq j\leq k} q_h^{\pi_j}(s,a)}. \quad (51)$$

The last inequality is because $(s,a) \in L_{k_{s,a,h},h}$, by the definition of $L_{k,h}$ in Eq. (42), we have

$$\frac{1}{4}\sum_{j<k_{s,a,h}} q_h^{\pi_j}(s,a) \geq \ln\frac{SAH}{\delta'}+1 \geq 1.$$

Define functions

$$G_{s,a,h}(x) := (x-\lfloor x\rfloor)\cdot q_h^{\pi_{\lceil x\rceil}}(s,a) + \sum_{k_{s,a,h}\leq j\leq\lfloor x\rfloor} q_h^{\pi_j}(s,a), \quad (52)$$

$$g_{s,a,h}(x) := q_h^{\pi_{\lceil x\rceil}}(s,a). \quad (53)$$

Roughly speaking, $G_{s,a,h}(x)$ is the linear interpolation of the sum of $q_h^{\pi_j}(s,a)$, and $g_{s,a,h}(x)$ is the step function whose steps are $q_h^{\pi_j}(s,a)$. We can easily check that when $x$ is not an integer,

$$\frac{\partial G_{s,a,h}(x)}{\partial x} = g_{s,a,h}(x). \tag{54}$$

Notice that the not-differentiable points (i.e., when $x$ is an integer) of $G_{s,a,h}(s,a)(x)$ are countable and will not affect the following calculation.

$$
\begin{aligned}
&\sum_{k=k_{s,a,h}}^{K} \frac{q_h^{\pi_k}(s,a)}{4 + \sum_{k_{s,a,h} \leq j \leq k} q_h^{\pi_j}(s,a)} \\
&= \int_{k_{s,a,h}-1}^{K} \frac{g_{s,a,h}(x)}{4 + G_{s,a,h}(\lceil x \rceil)} dx \quad \text{(by Eq. (52) and Eq. (53))} \\
&\leq \int_{k_{s,a,h}-1}^{K} \frac{g_{s,a,h}(x)}{4 + G_{s,a,h}(x)} dx \quad \text{(since } G_{s,a,h}(\cdot) \text{ is monotone increasing)} \\
&= \ln(4 + G_{s,a,h}(K)) - \ln(4 + G_{s,a,h}(k_{s,a,h}-1)) \quad \text{(by Eq. (54))} \\
&\leq \ln\left(4 + \sum_{k_{s,a,h} \leq j \leq K} q_h^{\pi_j}(s,a)\right) \\
&\leq \ln(4 + K). \tag{55}
\end{aligned}
$$

By Eq. (55), Eq. (50), and Eq. (51), the result of this lemma thus follows. □

## D    PROOF OF THEOREM 2

*Proof.* For the proof of PAC guarantee, we have

$$V_1^{*,F}(s_1) - V_1^{\pi^{\text{mix}}}(s_1) = V_1^{*,F}(s_1) - F\left(\left[\frac{1}{K}\sum_{k=1}^{K} V_{1,(i)}^{\pi_k}(s_1)\right]_{i=1,\cdots,N}\right).$$

Because $F$ is a concave function, by Jensen's inequality, we have

$$F\left(\left[\frac{1}{K}\sum_{k=1}^{K} V_{1,(i)}^{\pi_k}(s_1)\right]_{i=1,\cdots,N}\right) \geq \frac{1}{K}\sum_{i=1}^{K} F\left(\left[V_{1,(i)}^{\pi_k}(s_1)\right]_{i=1,\cdots,N}\right) = \frac{1}{K}\sum_{i=1}^{K} V_1^{\pi_k,F}(s_1).$$

Thus, we have $V_1^{*,F}(s_1) - V_1^{\pi^{\text{mix}}}(s_1) \leq \frac{1}{K}\sum_{k=1}^{K}\left(V_1^{*,F}(s_1) - V_1^{\pi_k,F}(s_1)\right) = \frac{\text{Reg}(K)}{K}$. By Theorem 1, we know that with high probability $\text{Reg}(K) = C_F \cdot \left(\tilde{\mathcal{O}}(H^2 N S\sqrt{AK}) + \tilde{\mathcal{O}}(HN^2 S^{3/2}A) + \tilde{\mathcal{O}}(H^2 N S^2 A)\right)$. Thus, by letting

$$\varepsilon = \frac{C_F \cdot \left(\tilde{\mathcal{O}}(H^2 N S\sqrt{AK}) + \tilde{\mathcal{O}}(HN^2 S^{3/2}A) + \tilde{\mathcal{O}}(H^2 N S^2 A)\right)}{K},$$

we can get

$$K = C_F \max\left\{\tilde{\mathcal{O}}(H^4 N^2 S^2 A/\varepsilon^2),\ \tilde{\mathcal{O}}(H^2 N^4 S^3 A^2/\varepsilon^2),\ \tilde{\mathcal{O}}(H^2 N^2 S^4 A^2/\varepsilon^2)\right\}.$$

Notice that for each episode we have $H$ samples. Thus, the total number of samples is

$$C = C_F \max\left\{\tilde{\mathcal{O}}(H^5 N^2 S^2 A/\varepsilon^2),\ \tilde{\mathcal{O}}(H^3 N^4 S^3 A^2/\varepsilon^2),\ \tilde{\mathcal{O}}(H^3 N^2 S^4 A^2/\varepsilon^2)\right\}.$$

The result thus follows. □

# E   PROOF OF THEOREM 3

Recalling the definition of suboptimality, we get Eq. (16). Term 2 of Eq. (16) is non-positive because $\hat{\pi}$ is the solution of Eq. (14).

We now bound Term 3. First, we specify the value of $b_h^r(s, a, \delta)$ and $b_h^p(s, a, s', \delta)$. For the offline setup, we denote $n_h(s, a, s')$ as the empirical value within the dataset. Now, set $b_h^r(s, a, \delta)$ as the value in (12) and $b_h^p(s, a, s', \delta)$ as the value in (32) respectively. From the Value-difference Lemma, for any $i$, we have

$$
V_{1,(i)}^{\hat{\pi}}(s, \underline{r}, \overline{p}) - V_{1,(i)}^{\hat{\pi}}(s, r, p) = \mathbb{E}_{p,\hat{\pi}} \left[ \sum_{h=1}^{H} (\underline{r}_{(i),h}(s_h, a_h) - r_{(i),h}(s_h, a_h)) | s_1 = s \right]
$$
$$
+ \mathbb{E}_{p,\hat{\pi}}[\sum_{h=1}^{H} \sum_{s'_{h+1}} (\overline{p}_h(s_h, a_h, s'_{h+1}) - p(s_h, a_h, a'_{h+1})) V_{h+1,(i)}]
$$
(56)

From Lemma 12, 13, and 14, we have $|\overline{r}_{(i),h}(s, a) - r_{(i),h}(s, a)| \leq b_h^r(s, a, \delta)$, and $|\overline{p}(s, a, s') - p(s, a, s')| \leq b_h^p(s, a, s')$ with probability $1 - \delta$. Since, $V_{h+1,(i)} \leq H$. Thus,

$$
V_{1,(i)}^{\hat{\pi}}(s, \underline{r}, \overline{p}) - V_{1,(i)}^{\hat{\pi}}(s, r, p) \leq
$$
$$
\mathbb{E}_{p,\hat{\pi}} \left[ \sum_{h=1}^{H} (\underline{r}_{(i),h}(s_h, a_h) - \overline{r}_{(i),h}(s_h, a_h) + b_h^r(s_h, a_h, \delta) + H \sum_{s'_{h+1}} b_h^p(s_h, a_h, s'_{h+1}, \delta)) \right]
$$
(57)

Now, by the definition of $\underline{r}$, we can bound the above by 0. Finally, using the fact that $F(\cdot)$ is monotone increasing, we can conclude that Term 3 is bounded by 0.

It remains to estimate Term 1. To that end, when the event $\mathcal{E}$ (defined in Definition 1) happens, we have

$$
\left| \underline{r}_{h,(i)}(s, a) - r_{h,(i)}(s, a) \right|
$$
$$
= \left| \overline{r}_{h,(i)}(s, a) - r_{h,(i)}(s, a) - b_h^r(s, a, \delta) - H \sum_{s' \in \mathcal{S}} b_h^p(s, a, s', \delta) \right| \text{ (by the definition of } \underline{r})
$$
$$
\leq 2 b_h^r(s, a, \delta) + H \sum_{s' \in \mathcal{S}} b_h^p(s, a, s', \delta) \text{ (by Definition 1 and the triangle inequality).}
$$
(58)

By Assumption 3, we have

$$
V_{h,(i)}^{\pi^*, F}(s, \underline{r}, \overline{p}) \in [\epsilon, H].
$$
(59)

Thus, we can apply Lemma 5. Specifically, under Assumption 3 and when the event $\mathcal{E}$ happens, we have

Term 1 of Eq. (16)

$$= V_1^{\pi^*, F}(s_1; r, p) - V_1^{\pi^*, F}(s_1; \underline{r}, \overline{p})$$

$$\leq C_F N \max_{i \in [N]} \left| V_{i,(i)}^{\pi^*}(s_1; r, p) - V_{1,(i)}^{\pi^*}(s_1; \underline{r}, \overline{p}) \right| \quad \text{(by Lemma 5)}$$

$$= C_F N \max_{i \in [N]} \left| \mathbb{E} \left[ \underline{r}_{h,(i)}(s_h, a_h) - r_{h,(i)}(s_h, a_h) \right] \right.$$

$$\left. + \mathbb{E} \left[ \sum_{h=1}^{H} \sum_{s' \in \mathcal{S}} \left( \overline{p}_h(s'|s_h, a_h) - p_h(s'|s_h, a_h) \right) V_{h+1,(i)}^{\pi^*}(s'; \underline{r}, \overline{p}) \right] \right| \quad \text{(by Lemma 10)}$$

$$\leq C_F N \max_{i \in [N]} \mathbb{E} \sum_{h=1}^{H} \left| \underline{r}_{h,(i)}(s_h, a_h) - r_{h,(i)}(s_h, a_h) \right|$$

$$+ C_F N \max_{i \in [N]} \sum_{h=1}^{H} \sum_{s' \in \mathcal{S}} \left| \left( \overline{p}_h(s'|s_h, a_h) - p_h(s'|s_h, a_h) \right) V_{h+1,(i)}^{\pi^*}(s'; \underline{r}, \overline{p}) \right|$$

(by the triangle inequality)

$$\leq C_F N \mathbb{E} \left[ \sum_{h=1}^{H} \left( 2 b_h^r(s_h, a_h, \delta) + H \sum_{s' \in \mathcal{S}} b_h^p(s_h, a_h, s', \delta) \right) \right]$$

$$+ C_F N \mathbb{E} \left[ \sum_{h=1}^{H} \sum_{s' \in \mathcal{S}} b_h^p(s_h, a_h, s', \delta) H \right] \quad \text{(by Eqs. (58) and (59) and Definition 1)}$$

$$= 2 C_F N \mathbb{E} \left[ \sum_{h=1}^{H} \left( b_h^r(s_h, a_h, \delta) + H \sum_{s' \in \mathcal{S}} b_h^p(s_h, a_h, s', \delta) \right) \right].$$

(The expectation $\mathbb{E}$ in the above equation is on the trajectories with optimal policy $\pi^*$ on the true MDP with $r$ and $p$.) The result of Theorem 3 thus follows.

# F    DETAILS OF FAIR ONLINE POLICY GRADIENT

The following proposition gives an estimation of the gradient based on the samples.

**Proposition 21.** *After collecting a set $\mathcal{D}$ of trajectories (with the policy $\pi_\theta$) where each trajectory $\tau \in \mathcal{D}$ contains the information $(s_h^\tau, a_h^\tau, r_h^\tau)_{h=1,2,\cdots,H}$, then $g \in \mathbb{R}^d$ is an unbiased[1] estimation of the gradient $\nabla_\theta V_1^{\pi_\theta, F}(s_1)$.*

$$g_{\text{max-min}} = \frac{1}{|\mathcal{D}|} \sum_{\tau \in \mathcal{D}} \sum_{h=1}^{H} R_{(\hat{i}_\theta^*)}(\tau) \nabla_\theta \log \pi_\theta(a_h^\tau|s_h^\tau), \text{ where } \hat{i}_{\theta_l}^* := \arg\min_{i \in [N]} \sum_{\tau \in \mathcal{D}} R_{(i)}(\tau).$$

$$g_{\text{proportional}} = \sum_{i=1}^{N} \frac{\sum_{\tau \in \mathcal{D}} \sum_{h=1}^{H} R_{(i)}(\tau) \nabla_\theta \log \pi_\theta(a_h^\tau|s_h^\tau)}{\sum_{\tau \in \mathcal{D}} R_{(i)}(\tau)}. \tag{60}$$

$$g_\alpha = |\mathcal{D}|^{\alpha-1} \sum_{i=1}^{N} \frac{\sum_{\tau \in \mathcal{D}} \sum_{h=1}^{H} R_{(i)}(\tau) \nabla_\theta \log \pi_\theta(a_h^\tau|s_h^\tau)}{\left( \sum_{\tau \in \mathcal{D}} R_{(i)}(\tau) \right)^\alpha}.$$

*Proof.* Based on the chain rule, we have the following results.

1. When $F = F_{\text{max-min}}$, let $i_\theta^* := \arg\min_{i \in [N]} V_{1,(i)}^{\pi_\theta, F}(s_1)$:

$$\nabla_\theta V_1^{\pi_\theta, F}(s_1) = \nabla_\theta V_{1,(i_\theta^*)}^{\pi_\theta}(s_1). \tag{61}$$

---

[1] An unbiased estimation means that when $|\mathcal{D}| \to \infty$, the estimated value approaches the true value.

Notice that $\nabla_{\boldsymbol{\theta}} \, i_{\boldsymbol{\theta}}^{*} = \mathbf{0}$ almost everywhere if $V_{1,(i)}^{\pi_{\boldsymbol{\theta}}, F}(s_1)$ is continuous w.r.t. $\boldsymbol{\theta}$.

2. When $F = F_{\text{proportional}}$:

$$\nabla_{\boldsymbol{\theta}} V_1^{\pi_{\boldsymbol{\theta}}, F}(s_1) = \nabla_{\boldsymbol{\theta}} \sum_{i=1}^{N} \log V_{1,(i)}^{\pi_{\boldsymbol{\theta}}}(s_1) = \sum_{i=1}^{N} \frac{\nabla_{\boldsymbol{\theta}} V_{1,(i)}^{\pi_{\boldsymbol{\theta}}}(s_1)}{V_{1,(i)}^{\pi_{\boldsymbol{\theta}}}(s_1)}. \tag{62}$$

3. When $F = F_{\alpha}$:

$$\nabla_{\boldsymbol{\theta}} V_1^{\pi_{\boldsymbol{\theta}}, F}(s_1) = \nabla_{\boldsymbol{\theta}} \sum_{i=1}^{N} \frac{1}{1-\alpha} \left( V_{1,(i)}^{\pi_{\boldsymbol{\theta}}}(s_1) \right)^{1-\alpha} = \sum_{i=1}^{N} \left( V_{1,(i)}^{\pi_{\boldsymbol{\theta}}}(s_1) \right)^{-\alpha} \nabla_{\boldsymbol{\theta}} V_{1,(i)}^{\pi_{\boldsymbol{\theta}}}(s_1). \tag{63}$$

It remains to approximate $\nabla_{\boldsymbol{\theta}} V_{1,(i)}^{\pi_{\boldsymbol{\theta}}}(s_1)$ and $V_{1,(i)}^{\pi_{\boldsymbol{\theta}}}(s_1)$ in the above equations. To that end, noticing that $V_{1,(i)}^{\pi_{\boldsymbol{\theta}}}(s_1) = \mathbb{E}_{\tau} \, R_{(i)}(\tau)$, we can approximate $V_{1,(i)}^{\pi_{\boldsymbol{\theta}}}(s_1)$ by the empirical average of $R_{(i)}$, i.e.,

$$\frac{1}{|\mathcal{D}|} \sum_{\tau \in \mathcal{D}} R_{(i)}(\tau). \tag{64}$$

Before calculating $\nabla_{\boldsymbol{\theta}} V_{1,(i)}^{\pi_{\boldsymbol{\theta}}}(s_1)$, we first list some equations that will be used later.

1. Probability of a trajectory:

$$\Pr(\tau|\boldsymbol{\theta}) = \prod_{h=1}^{H} p_h(s_{h+1}|s_h, a_h) \pi_{\boldsymbol{\theta}}(a_h|s_h). \tag{65}$$

2. The log-derivative trick:

$$\nabla_{\boldsymbol{\theta}} \Pr(\tau|\boldsymbol{\theta}) = \Pr(\tau|\boldsymbol{\theta}) \cdot \nabla_{\boldsymbol{\theta}} \log \Pr(\tau|\boldsymbol{\theta}). \tag{66}$$

3. Log-probability of a trajectory:

$$\begin{aligned}
\log \Pr(\tau|\boldsymbol{\theta}) &= \log \prod_{h=1}^{H} p_h(s_{h+1}|s_h, a_h) \pi_{\boldsymbol{\theta}}(a_h|s_h) \quad \text{(by Eq. (65))} \\
&= \sum_{h=1}^{H} \log p_h(s_{h+1}|s_h, a_h) + \log \pi_{\boldsymbol{\theta}}(a_h|s_h).
\end{aligned}$$

Thus, we have

$$\nabla_{\theta} \log \Pr(\tau|\theta) = \sum_{h=1}^{H} \nabla_{\theta} \log \pi_{\theta}(a_h|s_h). \tag{67}$$

Notice that to get the above equation, we use the fact that the transition probability $p$ is irrelevant to $\boldsymbol{\theta}$.

Now we are ready to calculate $\nabla_{\boldsymbol{\theta}} V_{1,(i)}^{\pi_{\boldsymbol{\theta}}}(s_1)$. We have

$$
\begin{aligned}
\nabla_{\boldsymbol{\theta}} V_{1,(i)}^{\pi_{\boldsymbol{\theta}}}(s_1) =& \nabla_{\boldsymbol{\theta}} \mathbb{E}_{\tau} R_{(i)}(\tau) \\
=& \nabla_{\boldsymbol{\theta}} \int_{\tau} \Pr(\tau|\boldsymbol{\theta}) R_{(i)}(\tau) \\
=& \int_{\tau} \nabla_{\boldsymbol{\theta}} \Pr(\tau|\boldsymbol{\theta}) R_{(i)}(\tau) \\
=& \int_{\tau} \Pr(\tau|\theta) \nabla_{\boldsymbol{\theta}} \log(\Pr(\tau|\boldsymbol{\theta})) R_{(i)}(\tau) \;\; \text{(by Eq. (66))} \\
=& \mathbb{E}_{\tau} \nabla_{\boldsymbol{\theta}} \log(\Pr(\tau|\boldsymbol{\theta})) R_{(i)}(\tau) \\
=& \mathbb{E}_{\tau} R_{(i)}(\tau) \nabla_{\boldsymbol{\theta}} \log \Pr(\tau|\boldsymbol{\theta}) \\
=& \mathbb{E}_{\tau} R_{(i)}(\tau) \sum_{h=1}^{H} \nabla_{\boldsymbol{\theta}} \log \pi_{\boldsymbol{\theta}}(a_h|s_h) \;\; \text{(by Eq. (67))}.
\end{aligned}
$$

Thus, we can approximate $\nabla_{\boldsymbol{\theta}} V_{1,(i)}^{\pi_{\boldsymbol{\theta}}}(s_1)$ by the following empirical average:

$$
\frac{1}{|\mathcal{D}|} \sum_{\tau \in \mathcal{D}} R_{(i)}(\tau) \sum_{h=1}^{H} \nabla_{\boldsymbol{\theta}} \log \pi_{\boldsymbol{\theta}}(a_h|s_h). \tag{68}
$$

The result of this proposition thus follows by plugging the empirical estimation Eqs. (64) and (68) into Eqs. (61) to (63). □

As an example, we show the whole algorithm for max-min fairness in Algorithm 2.

---

**Algorithm 2** Policy Gradient for Max-Min Fairness

1: **Initialize:** $\boldsymbol{\theta}_0 \in \mathbb{R}^d$, step size $\alpha' > 0$.
2: **for** each iteration $l = 0, 1, 2, \cdots,$ **do**
3:     Collect a set of trajectory $\mathcal{D}$ by using $\pi_{\boldsymbol{\theta}_l}$ where each trajectory $\tau \in \mathcal{D}$ contains the information $(s_h, a_h, \boldsymbol{r}_h)_{h=1,2,\cdots,H}$.
4:     For each collected trajectory, calculate its total reward for each agent
5:     For each agent $i$, get an estimation of its own value function $\hat{V}_{1,(i)}^{\boldsymbol{\theta}_l}(s_1) \leftarrow \frac{1}{|\mathcal{D}|} \sum_{\tau \in \mathcal{D}} R_{(i)}(\tau)$.
6:     Select the agent with the minimum estimated value $\hat{i}_{\boldsymbol{\theta}_l}^* \leftarrow \arg\min_{i \in \{1,2,\cdots,N\}} \hat{V}_{1,(i)}^{\boldsymbol{\theta}_l}(s_1)$.
7:     Calculate the estimated gradient $\boldsymbol{g} \in \mathbb{R}^d$ by

$$
\boldsymbol{g} \leftarrow \frac{1}{|\mathcal{D}|} \sum_{\tau \in \mathcal{D}} R_{(\hat{i}_{\boldsymbol{\theta}_l}^*)}(\tau) \sum_{h=1}^{H} \nabla_{\boldsymbol{\theta}_l} \log \pi_{\boldsymbol{\theta}_l}(a_h|s_h).
$$

8:     Update the parameters $\boldsymbol{\theta}_{l+1} \leftarrow \boldsymbol{\theta}_l + \alpha' \boldsymbol{g}$.
9: **end for**

---

## G    SIMULATION RESULTS

In Fig. 1, we plot the curves of $V_1^F(s_1)$ of the optimal policy (dashed red curves) and the curves of the policy calculated by Algorithm 1 (the blue curves) for different fair objectives. We can see that for all three different fair objectives, the solution of Algorithm 1 becomes very close to the optimal one after $K \geq 550$. This validates our theoretical result that the regret scales sub-linearly ($\tilde{\mathcal{O}}(\sqrt{K})$) since the average regret $\tilde{\mathcal{O}}(\frac{\sqrt{K}}{K})$ goes to zero when $K$ becomes larger.

In Fig. 2, we plot the curves of $V_1^F(s_1)$ of the optimal policy (dashed red curves) and the curves of the policy calculated by the policy gradient method (the blue curves). We use a two-layer fully-connected neural network with ReLU (rectified linear unit) as the policy model. During each iteration of the policy gradient algorithm, 20 trajectories are generated and collected under the current

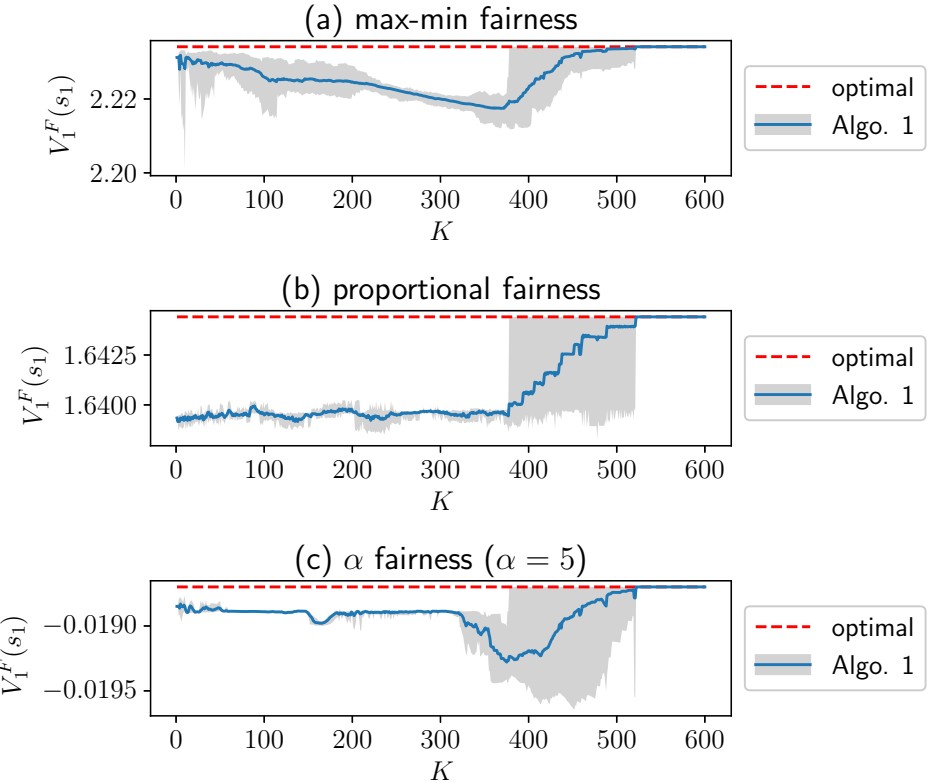

Figure 1: Curves of $V_1^F(s_1)$ of Algorithm 1 w.r.t. $K$ for different fair objectives. The solid blue curve is the average of 10 runs with different seeds. The shaded part denotes the range of these 10 runs (i.e., the range between the max and the min value).

policy. As shown by Fig. 2, such a policy gradient method can achieve the nearly optimal solution within 1000 iterations.

In Figs. 3 and 4, each point of the offline curve is calculated by applying the offline algorithm with the data generated by Algorithm 1 at $K$-th epoch. We can see that the offline policy is better than the online policy, which is reasonable because the offline policy only needs exploitation (i.e., choose the current best action), while the online policy needs to explore (i.e., try suboptimal actions to estimate the environment).

Figs. 5 and 6 show the cost of fairness. In Fig. 6, the fair optimal solution is evaluated in the classical objective (i.e., the sum of individuals' return), compared with the optimal classical (unfair) solution. We can see the gap is not very large, which suggests that the cost of fairness is relatively small. In contrast, in Fig. 5, the optimal classical (unfair) solution is evaluated in the fairness objective, compared with the optimal fair solution. We can see for some points the gap is significant, which justifies the necessity of a fair solution.

### G.1 CONFIGURATIONS OF SIMULATIONS

We use a synthetic MDP. Each term of the transition probability $p$ is *i.i.d.* uniformly generated between $[0, 1]$, and then we normalize $p$ to make sure that $\sum_{s' \in \mathcal{S}} p(s, a, s') = 1$. Every term of the true immediate reward $r$ is *i.i.d.* uniformly generated between $[0.15, 0.95]$. Each noisy observation of an immediate reward is drawn from a uniform distribution centered at its true value within the range of $\pm 0.05$ (thus all noisy observations are in $[0.1, 1]$). Figs. 1 and 2 use $S = A = N = 2$ and $H = 3$. Fig. 3 uses $A = 3, S = 3, N = 3, H = 4$. Fig. 4 uses $A = 2, S = 2, N = 3, H = 10$. Figs. 5 and 6 use $A = 2, S = 2, H = 3$.

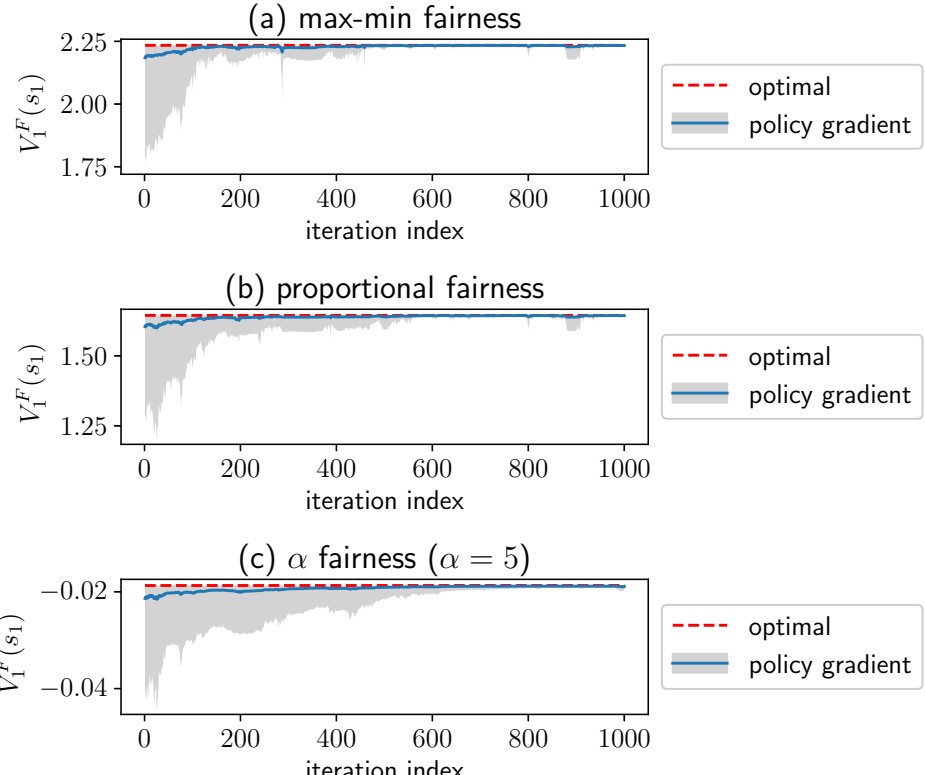

Figure 2: Curves of $V_1^F(s_1)$ of policy gradient w.r.t. the number of iterations for different fair objectives. The solid blue curve is the average of 10 runs with different seeds. The shaded part denotes the range of these 10 runs (i.e., the range between the max and the min value).

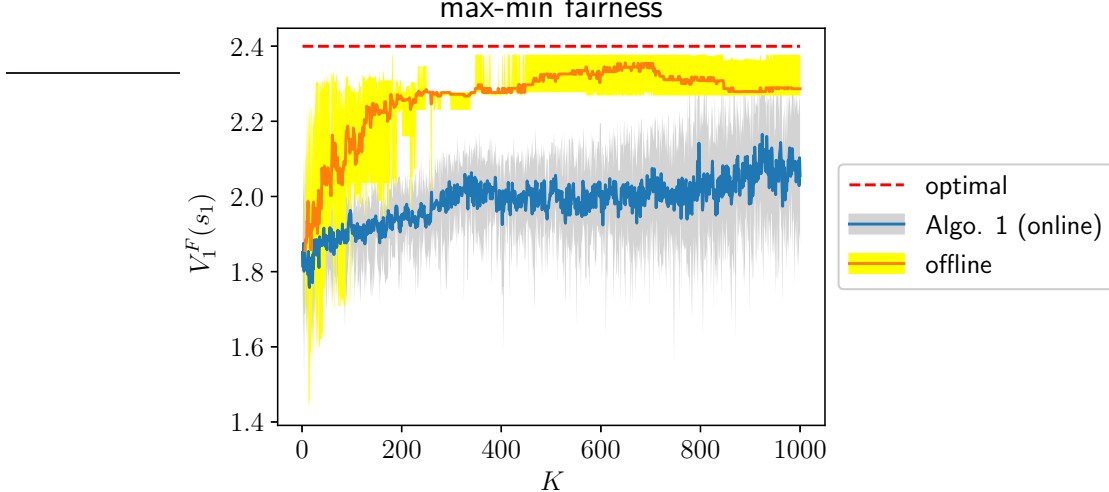

Figure 3: The curve of the offline algorithm performance w.r.t. the number of data, where the offline data is generated by the Algorithm 1 during the online learning process. Each point is the average of 10 random runs. ($A = 3, S = 3, N = 3, H = 4$)

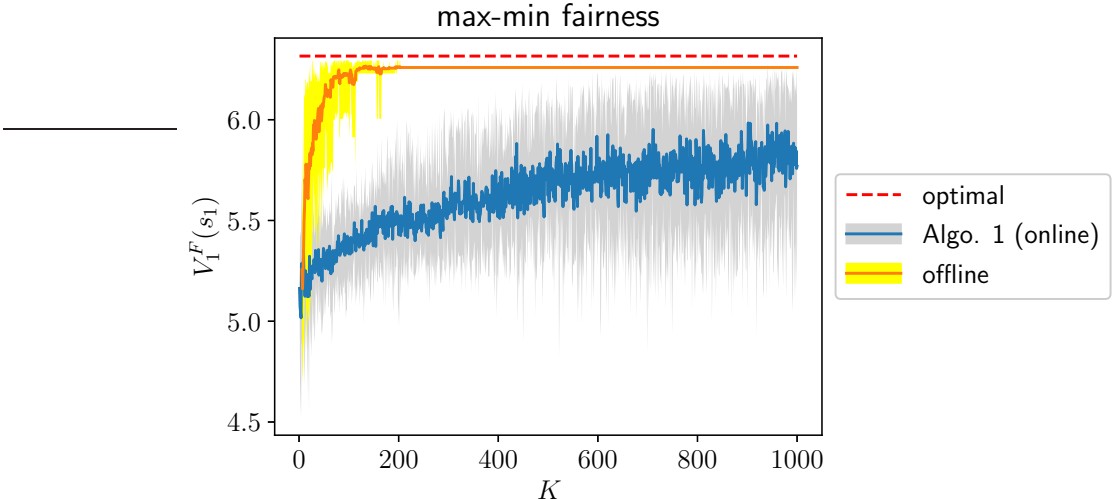

Figure 4: The curve of the offline algorithm performance w.r.t. the number of data, where the offline data is generated by the Algorithm 1 during the online learning process. Each point is the average of 10 random runs. ($A = 2, S = 2, N = 3, H = 10$)

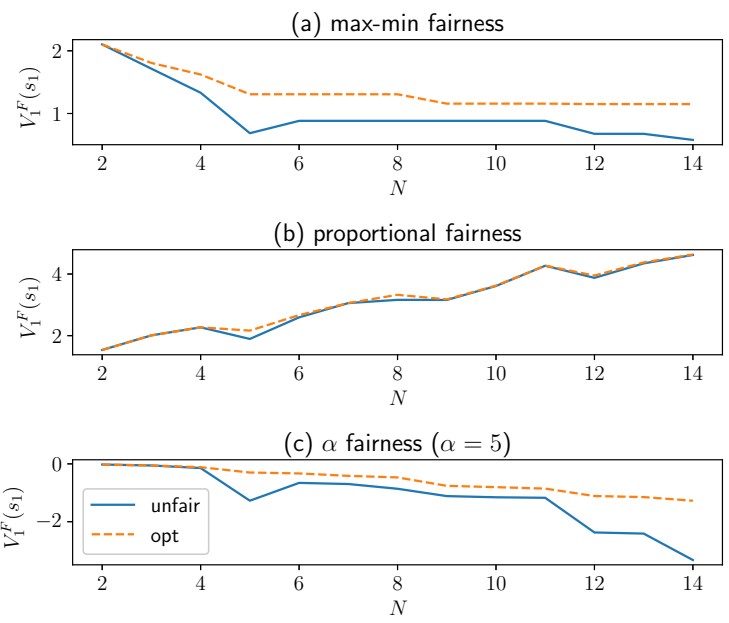

Figure 5: The value of the fairness objective for the optimal unfair (classical) policy

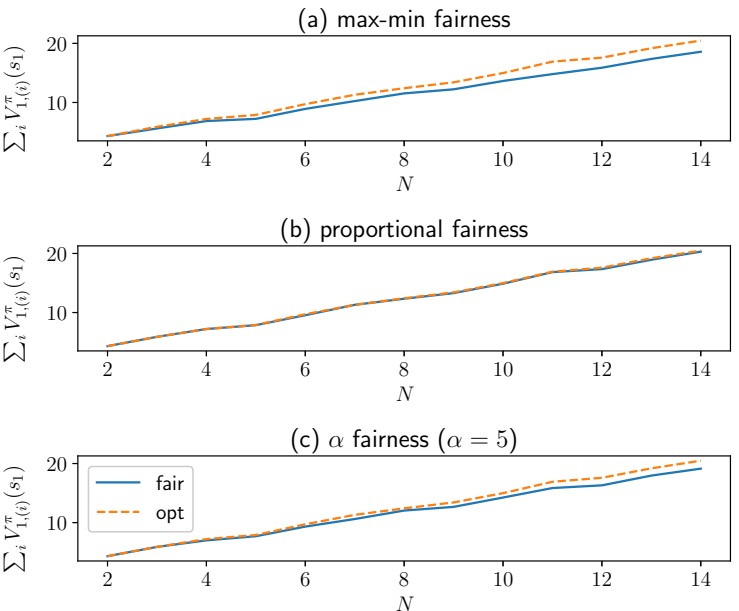

Figure 6: The value of the classical objective for the optimal fair policy.

