# OpenReview forum: "Achieving Fairness in Multi-Agent MDP Using Reinforcement Learning"
_ICLR.cc/2024/Conference — ICLR 2024 poster_

### Official Review · Reviewer_TT7q · 2023-10-26

**Soundness:** 3 good
**Presentation:** 3 good
**Contribution:** 3 good
**Rating:** 8
**Confidence:** 3

**Summary:**

This paper proposes a fairness function for multi-agent RL that ensures equitable rewards across agents. The authors then propose two algorithms for obtaining the optimal policy; first, they study the online RL setting in which they show that their method achieves sublinear regret in terms of number of episodes. Furthermore, they discuss a policy gradient variant of this method. Second, they propose an offline RL algorithm and bound its optimality gap. Experimental results are carried out on a very limited toy problem.

**Strengths:**

1. The paper is well-motivated and studies an important problem in RL.
2. The authors provide theoretical analysis of two algorithms for online and offline RL. Although I have not checked the proofs carefully.

**Weaknesses:**

1. The experimental results are very limited and cast doubts on the applicability of the proposed algorithms:
* The toy MDPs are very small and have a short horizon.
* The method is not compared with any other approach from the literature on fair MARL. There is basically no baseline in the experimental results.
* Since the authors are talking about resource allocation throughout the paper and that problem setting seems to be the main goal. I was surprised to see the experiments are on some randomly generated toy MDPs.
* I would have preferred to see the experimental results in the main body of the paper, perhaps instead of the proof outlines.

2. I find the paper hard to follow, and I think both the writing and the outline of the paper needs some improvements for more clarity and coherence. Some examples are:
* Some of the notation is cluttered and overloaded (e.g., equations 10-13) which makes the math hard to understand.
* While I appreciate the existence of proof outlines, in their current form they are vague (at least to me). Also, I am confused by their positioning in the text; for example, proof outline of Theorem 1 is not immediately after the Theorem.
* Some section titles are too generic without any useful information. For example Section 5 is titled “Algorithm” whereas Section 6 is titled “Offline Fair MARL”. It makes sense to rename Section 5 to something more informative (e.g., Online Fair MARL).

**Questions:**

1. Why is the horizon extremely short (H=3) in the experiments in Appendix G?
2. In Figure 3, why is the offline RL algorithm significantly outperforming the online RL algorithm? I find this a bit surprising because usually online only performs better and more importantly, your proposed online RL algorithm is optimistic whereas your offline RL algorithm is pessimistic.

---

> ### Author Response · Authors · 2023-11-22
> **Response to Reviewer TT7q**
>
> > The experimental results are very limited and cast doubts on the applicability of the proposed algorithms:
> The toy MDPs are very small and have a short horizon.
>
> **Response**: We added a new simulation figure with a longer horizon (H=10) in the new Figure 4 of the current revision, and we will add more simulations for the final version of this paper.
>
> ---
>
> > The method is not compared with any other approach from the literature on fair MARL. There is basically no baseline in the experimental results.
>
> **Response**: Fair MARL is a relatively new topic and that is why when we write the paper, there are almost no popular existing algorithms with regret guarantees that are suitable for a baseline in our setup (e.g., some algorithms have much higher computation complexity than ours).
>
> ---
>
> > Since the authors are talking about resource allocation throughout the paper and that problem setting seems to be the main goal. I was surprised to see the experiments are on some randomly generated toy MDPs.
>
> **Response**: Thanks for your suggestions. Our current simulation is on randomly generated toy MDPs because we want to speed up the simulation. Nonetheless, we agree with the reviewer that showing experiments on real resource allocation is more helpful. Although the rebuttal time period is too short to run simulations in this new setup, we will add these simulations on resource allocation in the final version.
>
> ---
>
> > I would have preferred to see the experimental results in the main body of the paper, perhaps instead of the proof outlines.
>
> **Response**: Thanks for your suggestion. In the final version, we plan to allocate some space (by shrinking the length of some other parts) in the main body to include experimental results.
>
> ---
>
> > I find the paper hard to follow, and I think both the writing and the outline of the paper needs some improvements for more clarity and coherence. Some examples are:
> Some of the notation is cluttered and overloaded (e.g., equations 10-13) which makes the math hard to understand.
>
> **Response**: Thank you for your comment. Due to space limitations, we had to put some less important math details in Appendix C. We will revise the paper further to ensure that the main body is self-contained.
>
> ---
>
> > While I appreciate the existence of proof outlines, in their current form they are vague (at least to me).
>
> **Response**: Due to the limit of the paper length, we use the proof outline only to show readers a rough idea of our proving methods. If the reviewer wants to know some specific parts of the proof outline, we are happy to provide more details. The full rigorous proof is in the appendix.
>
> ---
>
> > Also, I am confused by their positioning in the text; for example, proof outline of Theorem 1 is not immediately after the Theorem.
> Some section titles are too generic without any useful information. For example Section 5 is titled “Algorithm” whereas Section 6 is titled “Offline Fair MARL”. It makes sense to rename Section 5 to something more informative (e.g., Online Fair MARL).
>
> **Response**: Thanks for your suggestions. In the revision, we have changed the location of the proof outline and the section title as you suggested.
>
> ---
>
> > Why is the horizon extremely short (H=3) in the experiments in Appendix G?
>
> **Response**: We use a short horizon to speed up the simulation. To address the concern of the horizon length, in the revision, we have added experiments of a slightly longer horizon (H=10) in the new Fig. 4.
>
> ---
>
> > In Figure 3, why is the offline RL algorithm significantly outperforming the online RL algorithm? I find this a bit surprising because usually online only performs better and more importantly, your proposed online RL algorithm is optimistic whereas your offline RL algorithm is pessimistic.
>
> **Response**: The reason is that in Fig 3, for each $K$, the offline RL algorithm uses the same data collected by the online algorithm. In other words, the high-quality data explored by the online algorithm is also used as the input of the offline RL algorithm. Thus, the offline algorithm gets the full benefits of the high-quality data collected by the online algorithm. Meanwhile, the online algorithm needs to balance between the exploitation of the collected data and future exploration, while the offline algorithm totally works on exploitation. This is why the offline RL algorithm works better than the online algorithm in Fig. 3. We also agree that if the offline algorithm uses different data (e.g., randomly generated data), then the online algorithm may perform better than the offline one when the offline RL algorithm does not get access to the high-quality data collected by online RL.
>
> ---

---

> > ### Comment · Reviewer_TT7q · 2023-11-23
> >
> > I thank the authors for their reply. My concerns were adequately addressed and after reading other reviews, I have decided to increase my score.

---

### Official Review · Reviewer_KfsC · 2023-10-28

**Soundness:** 3 good
**Presentation:** 3 good
**Contribution:** 4 excellent
**Rating:** 8
**Confidence:** 3

**Summary:**

This paper studies fair MARL. It proposes online and offline RL algorithms and their deep policy gradient counterparts to maximize any fair social welfare from the alpha-fairness class in a multi-agent MDP. To this end, the paper notices that the Bellman operator cannot be readily applied when fairness is a concern, and proposes an alternative based on the occupancy measure. The regret bounds for the proposed algorithms are derived.

**Strengths:**

- MARL is relevant to the conference; fairness is relevant to MARL.
- The paper provides an impressive amount of theory. It also explores online and offline settings and extends RL algorithms to deep counterparts. The contributions listed in the Our Contributions section are original and significant.
- The theory is complemented by simple experiments.

**Weaknesses:**

Despite the vast contributions, I think the paper somewhat overclaims. I suggest to rewrite some parts of the Abstract, Introduction, and Related work. I elaborate below

- Fairness in (deep) RL, and in particular alpha-fairness, has been studied in multiple prior works. The Related Work section is too concise and does not give the existing literature enough merit. Example: Zimmer et al. 2021 is cited, but is claimed to focus on gini fairness. This is not true: they focus on a general class of fair SW functions, including alpha-fairness, and they have long-term fairness, and they propose a policy gradient algorithm. Another example: Ivanov et al. 2021 is missing, and they have long-term max-min fairness.
- In abstract, “the exploration of fairness in such systems for unknown environments remains open”. Yet many papers from section 2 are in model-free setting.
- The experiments are simple. I am not sure how to express it: they are as simple as they could be. One would actually have trouble to come up with simpler experiments. Two to four states, actions, agents, and transitions in an episode, all uniformly generated. This is ok by itself for a theoretical paper: after all, simple experiments are better than no experiments. But the authors claim that “... we also developed a policy-gradient-based algorithm that is applicable to large state space as well.” I would not call four a large number, so there is no evidence for this claim.

Other than these problems, the contribution is primarily theoretical and I only skimmed the theory. I lower my confidence score to reflect this.

**Questions:**

- Given my comment about the relation to literature, would it perhaps be more fair (pun not intended) to frame the paper as theoretically justifying existing approaches and proposing their new implementations rather than proposing new approaches?
- End of 5.1: why is the policy defined stochastically instead of deterministic argmax_a q? Wouldn’t the deterministic policy maximize social welfare?
- Isn’t the online algorithm in 5.2 a variation/extension of UCB for bandits?

---

> ### Author Response · Authors · 2023-11-22
> **Response to Reviewer KfsC (part 1)**
>
> > Despite the vast contributions, I think the paper somewhat overclaims. I suggest to rewrite some parts of the Abstract, Introduction, and Related work. I elaborate below:
> Fairness in (deep) RL, and in particular alpha-fairness, has been studied in multiple prior works. The Related Work section is too concise and does not give the existing literature enough merit. Example: Zimmer et al. 2021 is cited, but is claimed to focus on gini fairness. This is not true: they focus on a general class of fair SW functions, including alpha-fairness, and they have long-term fairness, and they propose a policy gradient algorithm. Another example: Ivanov et al. 2021 is missing, and they have long-term max-min fairness.
>
> **Response**: Thank you for your suggestion. We have now modified our claim. We agree that Zimmer et al. 2021 indeed considered alpha-fairness. However, unlike those previous works, we provide regret bounds. Further, unlike all the other papers, we also consider an offline RL setup.  We have now added the above in the paper.
> We did not find Ivanov et al. 2021. If the reviewer can provide the title or the link of the paper, we are happy to cite and compare with it in the final version.
>
> ---
>
> > In abstract, “the exploration of fairness in such systems for unknown environments remains open”. Yet many papers from section 2 are in model-free setting.
>
> **Response**: While we agree that there are model-free algorithms with good empirical performance, these algorithms do not provide theoretical guarantees, e.g., none of the algorithms provide a regret bound. This is what we meant in the above sentence. In the revision, we modified the sentence as the following: “...provably efficient exploration of fairness in such systems for unknown environments remains open”.
>
> ---
>
> > The experiments are simple. I am not sure how to express it: they are as simple as they could be. One would actually have trouble to come up with simpler experiments. Two to four states, actions, agents, and transitions in an episode, all uniformly generated. This is ok by itself for a theoretical paper: after all, simple experiments are better than no experiments. But the authors claim that “... we also developed a policy-gradient-based algorithm that is applicable to large state space as well.” I would not call four a large number, so there is no evidence for this claim.
> Other than these problems, the contribution is primarily theoretical and I only skimmed the theory.
>
> **Response**: Thanks for your understanding. In the current revision, we added a new simulation figure with a longer horizon (H=10) in the new Figure 4. In the final version, we will add more simulations.
>
> ---
>
> > Given my comment about the relation to literature, would it perhaps be more fair (pun not intended) to frame the paper as theoretically justifying existing approaches and proposing their new implementations rather than proposing new approaches?
>
> **Response**: Thank you for the interesting question. Note that our proposed algorithm is indeed a new algorithm that can achieve regret bound on fairness. Further, to the best of our knowledge, we propose the first offline RL algorithm with provable sub-optimal guarantee. As the reviewer pointed out there are algorithms that have attained good empirical performance, however, they do not provide any theoretical guarantees. We hope that our work can influence others to provide theoretical performance bounds for those approaches as well.
>
> ---
>
> > End of 5.1: why is the policy defined stochastically instead of deterministic argmax_a q? Wouldn’t the deterministic policy maximize social welfare?
>
> **Response**:  The last equation of Section 5.1 is due to the definition of occupancy measure $q$ in Eq. (8). The value of $q$ in the last equation of Section 5 is from the optimization result of Eq. (10). Then we just use Eq. (8) to find a policy that will lead to this occupancy measure, i.e., the last equation of Sec 5.1. Notice that after we get $q$, the optimization process is finished. Finding the policy that leads to this $q$ based on Eq. (8) does not involve any optimization.
>
> A deterministic policy is a special case of our solution. If the optimal policy is a deterministic policy, then for a given $s$, $q_h(s, a)$ will only have one non-zero (equal to 1) term among all $a$. However, since the Bellman equation does not apply in our fairness setup, the optimal fair policy may not necessarily be a deterministic policy.
>
> ---
>
> **Followed by "Response to Reviewer KfsC (part 2)"**

---

> > ### Author Response · Authors · 2023-11-22
> > **Response to Reviewer KfsC (part 2)**
> >
> > **Continued from "Response to Reviewer KfsC (part 1)"**
> >
> > > Isn’t the online algorithm in 5.2 a variation/extension of UCB for bandits?
> >
> > **Response**: We agree that Algorithm 1 seems very similar to UCB for bandits. However, there are subtle differences.  First, we want to emphasize that the construction of the “arms” and the confidence interval require specific consideration of the fairness objective. For example, we have to use the state-action-next-state occupancy measure as “arms” instead of the common state-action occupancy measure. Second, we consider an RL setting so the transition probability affects the performance, which is different from a classical UCB for bandits. In particular, we have to find the confidence interval of the transition probability. Finally, we have to find the upper confidence bound on the individual value function by combining the above. Details can be found in Appendix C.1.
> >
> > ---

---

> > > ### Comment · Reviewer_KfsC · 2023-11-22
> > > **Answer to rebuttal**
> > >
> > > Sorry for not providing all references, I meant this paper https://arxiv.org/abs/2102.12307.
> > >
> > > Thank you for revising the claims.
> > >
> > > Thank you also for answering my questions. The method seems quite intricate.
> > >
> > > Overall, I think fair MARL is important but understudied, especially theoretically. I skimmed other reviews and found concerns about practical applicability and assumptions being unrealistic, but I think the problem is vast and we gotta start somewhere. I will increase my score.

---

> > > > ### Author Response · Authors · 2023-11-22
> > > >
> > > > Thank you for providing the reference. We will include the reference in the final version. The paper  https://arxiv.org/abs/2102.12307 considers a mixture of individual value function and social welfare value function, which empirically shows good performance on achieving social welfare while training in a distributed fashion. In contrast, we provide regret bounds. Further, unlike all the other papers, we also consider an offline RL setup.
> > > >
> > > > We want to thank you for increasing the score and appreciating our contributions.

---

### Official Review · Reviewer_o4PJ · 2023-10-30

**Soundness:** 4 excellent
**Presentation:** 3 good
**Contribution:** 3 good
**Rating:** 8
**Confidence:** 3

**Summary:**

This paper aims to address the issue of fairness in multi-agent systems and introduces a fairness function that ensures equitable rewards across agents.
The authors propose an online convex optimization-based approach to obtain a policy constrained within a confidence region of the unknown environment.
Additionally, the authors demonstrate that their approach achieves sub-linear regret in terms of the number of episodes and provide a probably approximately correct (PAC) guarantee based on the obtained regret bound.
Furthermore, the authors propose an offline RL algorithm and bound the optimality gap concerning the optimal fair solution.

**Strengths:**

The paper is theoretically grounded and introduces fairness algorithms for both online and offline settings. In the online setting, the authors establish a regret bound with a Probably Approximately Correct (PAC) guarantee.
In the offline setting, the authors initially demonstrate the suboptimality of the policy.
Overall, the paper marks a promising start for bringing fairness into Multi-Agent Reinforcement Learning.

**Weaknesses:**

All concerns relate to the practicality of the method:

1. While the regret bounds are proven with a PAC guarantee, the cardinality of S is often huge in real-world settings, making it difficult to use.

2. Estimating the empirical average of p and r is often challenging in multi-agent settings, and the confidence interval may be unreliable.

3. The occupancy measure is based on the frequency of appearance for each state-action pair, which may be biased and difficult to count since we cannot always experience the entire environment.

**Questions:**

1.	Could you please provide some examples to demonstrate the effectiveness of fairness in a multi-agent system?
2.	I cannot understand what the experiment in the appendix demonstrates. Could you please provide a detailed description？(or how each agent can benefit from the fairness algorithm?)

---

> ### Author Response · Authors · 2023-11-22
> **Response to Reviewer o4PJ**
>
> > All concerns relate to the practicality of the method:
> While the regret bounds are proven with a PAC guarantee, the cardinality of S is often huge in real-world settings, making it difficult to use.
> The occupancy measure is based on the frequency of appearance for each state-action pair, which may be biased and difficult to count since we cannot always experience the entire environment.
>
> **Response**: We agree that the cardinality of S can be large in a multi-agent system and counting the state-action pair may be difficult in this case. However, in this paper, our focus is on dealing with fairness. We leave the possible state dimension reduction methods to future work. For example, one possible extension might be to consider linear MDP structure in [1-3]. For the concern that we cannot always experience the entire environment, our online algorithm indeed has an exploration term in the range of confidence intervals that puts priority on those unvisited state-action pairs.
>
> Also note that we use a policy-gradient approach which might be more suitable for large state-space as they can leverage on function approximation using neural networks. However, the sample complexity bound characterization of such approaches has been left for future investigations.
>
> [1]. Neu, Gergely, and Nneka Okolo. "Efficient global planning in large mdps via stochastic primal-dual optimization." In International Conference on Algorithmic Learning Theory, pp. 1101-1123. PMLR, 2023.
>
> [2]. Neu, Gergely, and Julia Olkhovskaya. "Online learning in MDPs with linear function approximation and bandit feedback." Advances in Neural Information Processing Systems 34 (2021): 10407-10417.
>
> [3]. Nachum, Ofir, and Bo Dai. "Reinforcement learning via fenchel-rockafellar duality." arXiv preprint arXiv:2001.01866 (2020).
>
>
>
> ---
>
> > Estimating the empirical average of p and r is often challenging in multi-agent settings, and the confidence interval may be unreliable.
>
> **Response**: We agree that in multi-agent settings, the empirical average of $p$ and $r$ needs a lot of samples. However, the definition of the confidence interval implies that it is an interval with high probability within which the true value lies. In our work, when there are few samples, the confidence interval will be large. When there are many samples, the confidence interval will be small. It is certainly true that such a confidence interval scales with the cardinality of the state-space. One possible solution might be to use a linear MDP structure where transition probability and the reward are linear in the known feature space in order to efficiently compute the confidence interval. The characterization of our policy for such a linear MDP constitutes an interesting future research direction.
>
> ---
>
> > Could you please provide some examples to demonstrate the effectiveness of fairness in a multi-agent system?
>
> **Response**: Certainly. The effectiveness of fairness is motivated by the literature on resource allocation in communication networking systems. For example, the max-min fairness is particularly effective when the overall system is limited by the agent/device that has the smallest utility. For example, the throughput over a route in a network (a set of links) is limited by the link with the smallest capacity. Thus, by maximizing the minimal agent’s (link’s) capacity, we effectively maximize the throughput of the network.
>
> ---
>
> > I cannot understand what the experiment in the appendix demonstrates. Could you please provide a detailed description? (or how each agent can benefit from the fairness algorithm?)
>
> **Response**: Thank you for this question. The fairness algorithm’s goal is to not necessarily benefit all agents, but to ensure that the disparity among agents is not very large. Usually, to achieve the fairness objective, some agents will need to sacrifice their rewards while some other agents will get more rewards (thus the result becomes “fair”). How the disparities are mitigated depends on the fairness objective. For example, in the max-min fairness, we tend to equalize the cumulative rewards achieved by each agent. In our simulations in the appendix, we consider all three types of fairness introduced in Section 3. For example, in Fig. 1(a), the blue curve approaches the red dashed line, which means that our Algorithm 1 eventually achieves the max-min fairness. In other words, the algorithm learned an optimal policy that can maximize the long-term return of the agent who has the smallest long-term return among all agents.
>
> ---

---

### Official Review · Reviewer_uKyX · 2023-10-31

**Soundness:** 2 fair
**Presentation:** 2 fair
**Contribution:** 2 fair
**Rating:** 3
**Confidence:** 4

**Summary:**

This paper explores the concept of fairness in multi-agent systems in unknown environments, casting the problem as a multi-agent Markov Decision Processes (MDPs) with a fairness-related objective function. The authors propose a Reinforcement Learning (RL) approach that maximizes fairness objectives such as Proportional Fairness and Maximin Fairness. When representing the policy in terms of the visitation measure, this paper shows that the optimal policy can be solved via convex optimization in the space of visitation measures. Based on this observation, a UCB-based online RL algorithm and a pessimism-based offline RL algorithm are proposed.

**Strengths:**

The fairness metric in multi-agent RL has not been extensively studied. This paper extends the UCB and Pessimism, two standard techniques of online and offline RL to this new problem.

**Weaknesses:**

1. Novelty. It seems that the main novelty of this paper is showing that the optimal fairness-aware policy can be solved by convex optimization in the space of visitation measure. Moreover, the function $F$ is Lipschitz and monotone in each coordinate. Based on this observation, one can easily extend the performance-difference lemma for standard MDP to this new problem.

2. The setting seems a bit restrictive. In particular, this paper considers the centralized setting and assumes all the $N$ agents have the same action $a$. This makes the problem no different from a single-agent MDP with $N$ different reward functions. Moreover, the assumption that the reward function is bounded from below by $\epsilon/ H$ seems an unnatural assumption.

3. It seems unclear whether the fairness notions studied in this work include most of the common fairness metrics. For example, existing works also study Generalized Gini Social Welfare Function in the context of MARL.

**Questions:**

1. Given existing works on UCB-based and pessimism-based RL, what are the technical novelties?

2. Can you run some toy-ish experiments to showcase the performance of the algorithm? In simulation experiments, how do we measure fairness-related regret?

---

> ### Author Response · Authors · 2023-11-22
> **Response to Reviewer uKyX (Part 1)**
>
> >It seems that the main novelty of this paper is showing that the optimal fairness-aware policy can be solved by convex optimization in the space of visitation measure. The function F is Lipschitz and monotone in each coordinate. One can easily extend the performance-difference lemma for standard MDP to this new problem.
>
> >Given existing works on UCB-based and pessimism-based RL, what are the technical novelties?
>
> **Response**: For the novelty of the results, to the best of our knowledge, our work is the first to give the regret/suboptimality bound for both online and offline fair RL algorithms. We achieve sub-linear regret bound and order (nearly) optimal in $K$.
>
> Regarding the technical novelty, although we acknowledge that certain components/steps of our methods (e.g., using confidence bound to quantify the uncertainty) may seem similar to some online convex work, our work, in general, is different. This is because the max-min, proportional, and alpha fairness objectives have not been studied in the RL literature, and the methods for a general concave objective do not fit here because the fairness objective (proportional and alpha fairness) is not always L-Lipchitz. This critical difference between the fairness objective and the general Lipschitiz continuous concave objective is also one of our motivations. Specifically, since the proportional and alpha fairness objective function can go to infinity and the regret can be unbounded, we have to make an additional requirement in Assumption 1 that the instant reward should not be too small. Related to this point, the technical challenge/novelty is how to connect this additional requirement to the performance of our algorithm. Indeed, as we show in the main results, this effect depends on the specific type of fairness, which is characterized by the coefficient $C_F$. In this regard, we state and prove Lemma 5 to show the dependency over different fairness functions, which we use to bound the regret.
>
> Furthermore, there are subtle differences in the algorithmic front as well. For example, we find that instead of a state-action occupancy measure, we need to consider a state-action-next state occupancy measure for efficient computation for an online approach. On the other hand, we have to modify the reward function in order to find the policy in the offline setup.
>
> ---
>
> > The setting seems a bit restrictive. This paper considers the centralized setting and assumes all the $N$ agents have the same action $a$. This makes the problem no different from a single-agent MDP with different reward functions. The assumption that the reward function is bounded from below by $\epsilon/H$ seems an unnatural assumption.
>
> **Response**: In our setting, we consider a centralized decision maker that learns the fair policy for all agents. Different agents can have different actions and different rewards, while the centralized decision-maker can view the actions of all agents as an aggregated action. However, the reward cannot be aggregated because of the fairness requirement. Therefore, the MDP is not equivalent to a single-agent MDP. While we consider that a central agent is taking decisions, our focus is on how those decisions can be fair to the agents.
>
> Further note that we can consider the setup where each individual agent is making a decision (here, the action-space can be different for different agents). Our algorithm is still applicable and it is equivalent to a centralized-training-and-decentralized-execution framework [1-2]. In particular, the optimal joint action can be learned by a centralized agent, then each agent can execute its own action based on the global state. The consideration of a decentralized training framework where each agent can learn only from its local information has been left for future investigations.
>
> [1]. Chen, Gang. "A New Framework for Multi-Agent Reinforcement Learning--Centralized Training and Exploration with Decentralized Execution via Policy Distillation." arXiv:1910.09152 (2019).
>
> [2]. Zimmer, Matthieu, Claire Glanois, Umer Siddique, and Paul Weng. "Learning fair policies in decentralized cooperative multi-agent reinforcement learning." ICML 2021.
>
> ---
>
> > It seems unclear whether the fairness notions studied in this work include most of the common fairness metrics. For example, existing works also study Generalized Gini Social Welfare Function in the context of MARL.
>
> **Response**: We agree that there are some other fairness types in the literature. However, in our paper, we have considered max-min, proportional, and alpha fairness, which are the most common/standard types in the literature of resource allocation (e.g., see Mo & Walrand (2000)).
>
> Jeonghoon Mo and Jean Walrand. Fair end-to-end window-based congestion control. IEEE/ACM Transactions on networking, 8(5):556–567, 2000.
>
> ---
>
> **Followed by "Response to Reviewer uKyX (Part 2)"**

---

> > ### Author Response · Authors · 2023-11-22
> > **Response to Reviewer uKyX (Part 2)**
> >
> > **Continued from "Response to Reviewer uKyX (Part 1)"**
> >
> > > Can you run some toy-ish experiments to showcase the performance of the algorithm? In simulation experiments, how do we measure fairness-related regret?
> >
> > **Response**: Thank you for your suggestion. Yes, we show simulation results in Appendix G. In all simulation figures, we plot the optimal fair policy (red horizontal dashed line). The regret is measured by the gap to this curve of the optimal fair policy. The empirical result also shows that the difference with the result achieved by our policy and the policy which is not fair (rather, it only maximizes the sum of the individual value functions).
> >
> > ---

---

> > ### Comment · Reviewer_uKyX · 2023-11-22
> >
> > I would like to thank the authors for trying to address my concerns, which still remain.
> >
> > **Technical novelty** I appreciate that the authors admit some of the techniques of regret analysis appear in the literature. To make this statement clearer, in particular, using confidence bound to quantify the uncertainty and bounding the regret using a pigeon-hole argument are widely used in prior works. The idea of regret decomposition is also standard, but I agree that decomposing a fairness-related regret is novel. It seems that the **regret decomposition of the fairness-related regret notion** is the only technical novelty.
> >
> > Let's think more deeply about this novelty. Looking at the regret analysis, the key step is Lemma 5, which shows that the three fairness objectives are Lipschitz with respect to the maximum of the value difference errors of each agent. Then, the value difference error is the same quantity that appears in the standard online RL literature so standard techniques can be applied -- regret decomposition via value difference lemma, pigeon-hole type regret bound, and uncertainty quantification.
> >
> > Thus, it seems that the main technical contribution beyond existing online RL literature is Lemma 5. As for Assumption 1, I don't think this is a technical contribution, rather, it is a **restriction** that makes the Lemma 5 go through. It also seems very strong because in many experiment environments (just consider OpenAI Gym), the rewards are sparse -- a lot of states have zero rewards.
> >
> >
> > **Extension to CTDE** seems standard to me. In terms of the analysis, we just need to call $a$ $(a^1, \ldots, a^N)$ and the same analysis holds. It would be fundamentally different if the training is also decentralized.

---

> > > ### Author Response · Authors · 2023-11-22
> > >
> > > > Concern about novelty, Assumption 1, and Lemma 5.
> > >
> > > **Response**: We thank the reviewer for acknowledging our technical novelty on the regret decomposition of the fairness-related regret notion. This is one of our main objectives in the online setup, i.e., we want to use RL to achieve a sub-linear regret bound with high probability and handle the technical difficulties that emerge in this process. Of course, we are not claiming that our contribution is towards creating a novel algorithm for very general online convex RL. Instead, we want to investigate and solve the difficulties when we apply the standard techniques to prove regret bounds when we have a fairness objective. This is exactly why we have Assumption 1 and Lemma 5, because standard techniques cannot achieve the regret bound without modification or restriction. We also want to mention that Assumption 1 is the requirement of the fairness objective itself instead of some artificially made assumption, since the regret of proportional fairness and $\alpha$ fairness can go to *infinity* in the worst case without Assumption 1, regardless of any RL algorithms and techniques.
> > >
> > > ---
> > >
> > > > It also seems very strong because in many experiment environments (just consider OpenAI Gym), the rewards are sparse -- a lot of states have zero rewards.
> > >
> > > **Response**: While we agree that in many environments the rewards can be sparse, what we are saying is that for some fairness metrics we need to lower bound the reward $\epsilon/H$ amount away from $0$. This can be done as post-processing where one can add $\epsilon/H$ if the reward returned by the environment as $0$. Note that for max-min fairness, we do not need such a requirement.
> > >
> > > While this assumption is necessary for our analysis in order to bound the function values in the worst case, in practice, even though some rewards are $0$, our algorithm will not choose those actions as it would get a negative infinite value function. Hence, this assumption may be relaxed for practical implementation like Open AI Gym.
> > >
> > > ---
> > >
> > > > Extension to CTDE seems standard to me. In terms of the analysis, we just need to call a = (a1 … aN) and the same analysis holds. It would be fundamentally different if the training is also decentralized.
> > >
> > > **Response**: As the reviewer suggested, indeed, our approach can be implemented in CTDE fashion which means that our algorithm can be executed (after the training phase) in decentralized fashion. We agree that it would be fundamentally different if the training is also decentralized. Indeed, this is an ongoing research topic even without fairness consideration.
> > >
> > > ---

---

### Meta-Review · Area_Chair_raGe · 2023-12-07

**Metareview:**

This paper analyses the theoretical properties of a MARL algorithm that accomplishes a measure of fairness in a multi-agent setting.

Strengths of the paper:
- The theoretical properties of fair algorithms in MARL have not been widely studied
- The content of the paper is correct according to the reviewers
- The paper is overall easy to follow

Weakness of the paper:
- The main weakness of the paper is limited novelty (see reviewer's uKyX comments)
- A further weakness is the very restricted nature of the analysis, a Multi-agent Finite Horizon MDP with lower bounded rewards.

On balance, I recommend this paper for acceptance if there is sufficient space but also request that the authors update and clarify their specific claims to novelty inline with the exchange of comments from reviewer uKyX.

**Justification For Why Not Higher Score:**

Limit novelty.

**Justification For Why Not Lower Score:**

On balance I think this paper will be useful for the community to build upon.
Having a clearly written and correct paper is not super common in the first place and novelty is more subjective.

---

### Decision · Program_Chairs · 2024-01-16

Accept (poster)